



# Spatio-temporal analysis of slope-type debris flow activity in Horlachtal, Austria based on orthophotos and LiDAR data since 1947

Jakob Rom[1], Florian Haas[1], Tobias Heckmann[1], Moritz Altmann[1], Fabian Fleischer[1], Camillo Ressl[2], Sarah Betz-Nutz[1], Michael Becht[1]

[1]Chair of Physical Geography, Catholic University of Eichstätt-Ingolstadt, 85072 Eichstätt, Germany
[2]Department of Geodesy and Geoinformation, Technische Universität Wien, 1040 Wien, Austria

*Correspondence to*: Jakob Rom (JRom@ku.de)

**Abstract.** In order to get a better understanding about the future development of alpine slope-type debris flows in the frame of climate change, complete and gapless records for the last century for this type of geomorphologic process are necessary. However, up to now such records are scarce. Here, the slope-type debris flow activity in the Horlachtal, Austria since 1947 is investigated with the help of historic and recent area-wide orthophotos. Using geomorphological mapping, both spatial and temporal variabilities in debris flow dynamics can be shown. The results indicate short-term variations rather than consistent increasing or decreasing trends of slope-type debris flow activity in Horlachtal. Specifically, three active periods between 1954 and 1973, 1990 and 2009 as well as between 2015 and 2018 can be registered. Analyses of the deposited debris flow volumes show that for parts of the study area the largest volumes appeared in the early 1990s which might have even influenced the dynamics in the following years. Studies on the spatial variabilities revealed differences of slope-type debris flow activity within the study area and point to local thunderstorms as triggers for debris flows. However, long-term precipitation data of high temporal resolution do not reveal increasing or decreasing tendencies in the number of such events.

## 1 Introduction

Debris flows are gravitational mass movements consisting of granular solids mixed with water that can reach high velocities (Varnes, 1978) and occur in mountainous regions around the world as natural hazards (Dowling and Santi, 2014). In high alpine regions, this process is of great importance for the sediment budget (Heckmann et al., 2012; Curry et al., 2006; Rainato et al., 2017; Hilger, 2017; Theule et al., 2012) as they couple sediment sources on slopes with alpine streams (Heckmann and Schwanghart, 2013; Iverson, 2012). Therefore, debris flows are a very important process in high alpine geomorphology and landscape evolution, as a change in debris flow activity has a high impact on sediment balances. In alpine environments, debris flows initiated in torrent beds (torrent bed type or channel-type) can be distinguished from debris flows initiated on slopes (slope-type or hillslope debris flows). These types not only differ in various geomorphic characteristics like flow length, drainage area or slope values (Chen et al., 2009), they also show different initiation mechanisms (Sassa, 1984).

Because of its importance for high alpine geosystems, there are many attempts trying to model debris flows in order to predict their appearance, velocities or ranges (Wichmann, 2017; Turnbull et al., 2015; Wu, 2015). But the changing environmental



parameters caused by climate change (Nogués-Bravo et al., 2007; Beniston, 2005, 2003) might have an impact on debris flow occurrence and properties, and must hence be accounted for in modelling effort. Therefore, in order to predict debris flow dynamics in the future, it is necessary to understand debris flow behaviour in the past and especially in the last decades that witnessed the most intense climatic changes. However, there are not many studies containing (near) complete debris flow

records beyond the last few decades in alpine catchments. Some authors used historical documents to reconstruct debris flow events (D'Agostino and Marchi, 2001; Tropeano and Turconi, 2004) but there are great uncertainties, especially for earlier periods (Marchi and Tecca, 2006). In addition, these archives often do not cover smaller debris flows and high-altitude regions far from settlements and focus primarily on channel-type debris flows. Other methods for reconstructing debris flow activities include dendrogeomorphology (Lopez Saez et al., 2011; Stoffel, 2010; Bollschweiler et al., 2008) or lichenometry (Helsen et

al., 2002; Innes, 1983). However these methods depend on the presence of suitable vegetation and are only partly useable above the treeline. The usage of area-wide LiDAR (Light Detection and Ranging) data as a basis for determining debris flow activity can reveal past debris flow events at higher resolution (Dietrich and Krautblatter, 2017; De Haas and Densmore, 2019) but the availability of these data is restricted to the last two decades and thus in a period, which is entirely dominated by climate change. Aerial images and orthophotos are used to detect changes in alpine environments and are available on an area-wide

basis back to ca. 1950 for most regions of the Alps (Bayle, 2020; Altmann et al., 2020; Fleischer et al., 2021). Because they cover whole catchments in great detail, historical and recent orthophotos are used to detect and date debris flow processes (Jomelli et al., 2007; Jomelli et al., 2003; Dietrich and Krautblatter, 2017).

A synthesis of previous studies on historical long-term development of debris flows shows that the results are not univocal. Dietrich and Krautblatter (2017) show an enhanced debris flow activity in Plansee, Austria, since the 1980s when investigating

the debris flow activity between 1947 and 2010. Other studies seem to confirm an increase in debris flow frequency when considering a long investigation period (Winter, 2020; Pelfini and Santilli, 2008; Kiefer et al., 2021). Due to changes in sediment supply, Hirschberg et al. (2021) stated that the number of debris flows in the Illgraben in Switzerland will decrease. However, several studies revealed no trends in debris flow activity besides of short-term fluctuations (Stoffel et al., 2014; Stoffel et al., 2005; Stoffel, 2010; Bollschweiler and Stoffel, 2010; Lopez Saez et al., 2011; Bollschweiler et al., 2008).

In this paper, the slope-type debris flow record in the Horlachtal in the central Alps of Austria between 1947 and 2020 is established using historical and recent orthophotos as well as LiDAR elevation models. The work is based on a precise mapping of all visible debris flows since 1947, which allows process frequencies to be derived. Process magnitudes are derived from LiDAR data as well as from an established area–volume relationship of debris flow deposits (Hilger, 2017; Larsen et al., 2010; Bennett et al., 2012). Thus obtained, the debris flow volumes are correlated with parameters of the respective hydrological

catchment areas in order to improve the understanding of spatial differences in debris flow activity. With the help of temporal high resolution precipitation data, this paper aims to analyse the spatial and temporal differences in slope-type debris flow activity in the Horlachtal, in order to gain a better understanding of the process behavior throughout the past seven decades and thus to link the result with changes in forcing due to climate change.





## 2 Study area

The Horlachtal study area is located in the northern part of the central Alps (Fig. 1) and forms a side valley of the Ötztal. The Horlachtal is drained by the Horlachbach, which flows as a hanging valley over the Stuibenfall waterfall into the main stream of the Ötztal (Ötz). The Horlachtal itself can be subdivided into three north-south striking tributary valleys (Grastal - GT, Larstigtal - LT and Zwieselbachtal - ZT), in addition to the east-west striking main valley (HT) as well as the tributary valleys Finstertal - FT and Weites Kar – WK (Tab. 1). The main outflow of the valley is captured by a gauging station in Niederthai,

which is located close to the area outlet at the Stuibenfall. Another gauging station is operated by the Tyrolean Hydropower Company (TIWAG) at Horlachalm, where part of the discharge is captured by a Tyrolean weir and fed to the Finstertal reservoir near Kühtai via underground tunnel systems in order to use it for hydropower. The Horlachtal spans elevations of 1557 m to 3340 m and shows a typical altitudinal alpine gradation of the vegetation. About 1.54 % of the area is currently glaciated, with the Grastalferner as the biggest glacier (ca. 0.48 km²) in the study area, whose outflow is buffered by the

Grastalsee. The Horlachtal shows the typical geomorphic process dynamics of high mountain regions, including rock glaciers in the upper areas, which testify to the presence of permafrost.

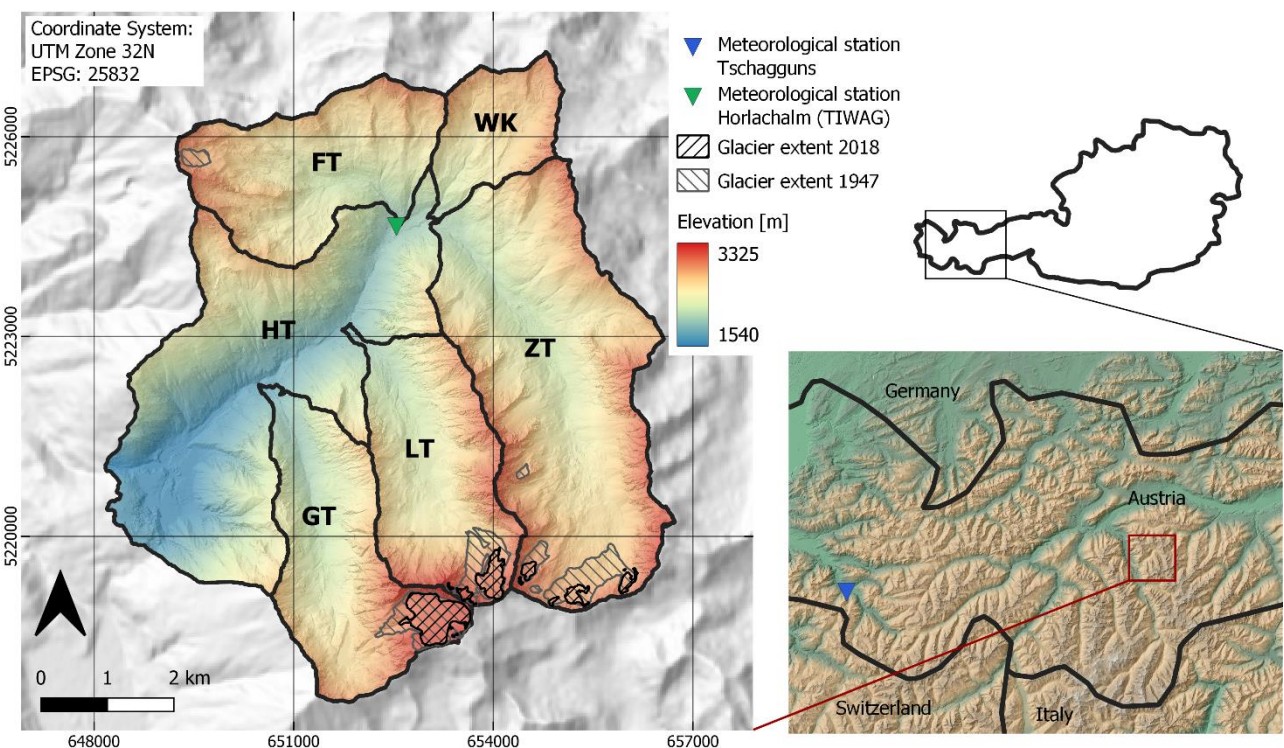

**Figure 1: Location of Horlachtal in the Stubai Alps. The study area is divided in the sub-catchments main valley (HT), Grastal (GT), Larstigtal (LT), Zwieselbachtal (ZT), Weites Kar (WK) and Finstertal (FT). Shown glacier extents were mapped based on orthophotos of the corresponding years. Elevation data of the study area are based on airborne LiDAR data from 2019. Large-scale**
**elevation data in the background are based on ALOS Global Digital Surface Model ©JAXA.**





Geologically, the study area is located in the Ötztal Massif with predominant gneisses and mica schists, which strike in an east-west direction parallel to the main valley (Geitner, 1999; Becht, 1995). Due to their tectonic history, the rocks are very susceptible to weathering, which leads to a high rockfall activity and ample availability of debris for debris flows, which led to the formation of in part very large debris cones.


**Table 2: Attributes of the different sub-catchments in the study area based on airborne LiDAR data of 2017 (Province of Tyrol).**

| sub-catchment | area [km²] | glaciated 2018 [%] | glaciated 1947 [%] | elevation mean [m] | elevation min [m] | elevation max [m] | elevation range [m] | slope mean [degree] |
|---|---|---|---|---|---|---|---|---|
| HT | 15.407 | 0.00 | 0.00 | 2092.9 | 1557.4 | 3000.5 | 1443.0 | 33.8 |
| GT | 7.386 | 6.48 | 10.08 | 2553.7 | 1702.3 | 3339.8 | 1637.5 | 35.6 |
| LT | 7.046 | 2.97 | 8.20 | 2621.4 | 1826.2 | 3339.8 | 1513.6 | 36.5 |
| ZT | 15.06 | 1.04 | 5.05 | 2618.3 | 2042.4 | 3240.9 | 1198.5 | 33.1 |
| WK | 3.042 | 0.00 | 0.00 | 2633.4 | 2050.1 | 3087.4 | 1037.2 | 29.6 |
| FT | 6.95 | 0.00 | 1.30 | 2569.4 | 1967.1 | 3060.5 | 1093.4 | 31.5 |
| Total | 54.891 | 1.54 | 4.00 | 2514.9 | 1557.4 | 3339.8 | 1782.4 | 33.3 |

Due to its location in the central Alps, the Horlachtal is protected from advective precipitation, so that the annual total of precipitation here is lower than for example in the Northern Alps (Geitner, 1999; Becht, 1995). The mean annual precipitation

between 1990 and 2019 add up to 817 mm, the mean annual temperature within the same time frame is 3.1 °C at the meteorological station Horlachalm (1910 m (all elevation data throughout this study refer to ellipsoid elevations); see Fig. 1 for the location within the study area; data courtesy of Tyrolean Hydropower Company TIWAG). Most precipitation occurs during the summer months and is primarily the result of convective precipitation during summer thunderstorms (Fig. 2).

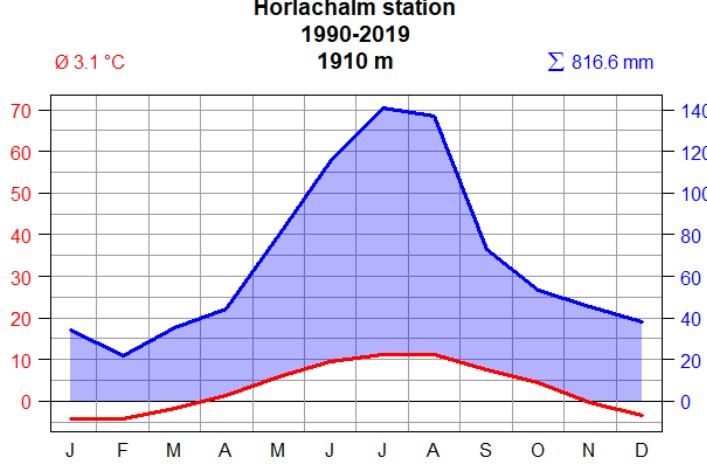

**Figure 2: Climate diagram of the Horlachalm station (1910 m) using temperature and precipitation data between 1990 and 2019.**
**Data source: TIWAG.**



In Horlachtal, slope-type debris flows occur primarily at the contact zone between steep bedrock and an adjacent talus slope. The catchments of the debris flows are developed in the steep bedrock sections. This type of debris flows can thus be defined as type 2 in Zimmermann (1990) or as type 1 in Wichmann (2006) and Rieger (1999). The influence of the morphometry of the hydrological catchment of a slope-type debris flow can be decisive with regard to its activity and magnitude (Becht and

Rieger, 1997; De Haas and Densmore, 2019; Dietrich and Krautblatter, 2019; Marchi et al., 2019; Shen et al., 2012; Wilford et al., 2004) and should be taken into account when analysing magnitudes of debris flows. The debris flows in Horlachtal occur in transport-limited hillslope systems and are triggered by precipitation events of about 20 mm in 30 min (Becht, 1995; Becht and Rieger, 1997).

## 3 Materials and methods

**3.1 Debris flow inventory using orthophotos**

The basis for all further evaluation methods was the multi temporal mapping of individual debris flows since 1947 in the whole study area. Debris flow inventories already existed especially in LT and ZT (Rieger, 1999; Thiel, 2013; Heckmann et al., 2014), which were carefully checked and updated for the whole study area using historical and recent orthophotos. All orthophotos used for this purpose and their characteristics are listed in Tab. 2. In some years, the aerial images do not cover

the entire study area. The missing regions were supplemented with aerial images from other flight campaigns with a temporal divergence of one to three years. The original aerial images of 1947, 1953, 1954, 1970 and 1973 were available from the archives of the Province of Tyrol. The scanned aerial images were oriented and calibrated in a bundle block adjustment (McGlone et al., 2004) using ground control points. These points were manually identified in recent data (orthophoto and digital elevation model) by looking for unique features (mostly rocks) in stable areas. After the bundle block adjustment, a

digital surface model was derived by means of image matching and used to create an orthophoto mosaic for the mentioned years. The remaining referenced orthophotos were taken from the web map service of the Province of Tyrol ([www.data.gv.at](www.data.gv.at)). Since the mapping is influenced by subjective interpretation of the orthophoto, it was done by one and the same person; visible typical debris flow deposits on the talus slopes were digitised into polygon shapefiles. In addition to the deposits, the starting zones in the aerial photographs were determined using the visible erosion areas. For debris flows with a hydrological catchment

in bedrock, these are primarily located in the direct transition from the bedrock area to the adjacent scree slopes (Rieger, 1999). If debris flow-typical process forms (transport channels, levées, deposits) have emerged during the comparison of two consecutive orthophotos, a new debris flow was mapped and dated to the time interval between the dates of image acquisition. The example in Fig. 3 shows debris flow landforms that emerged between 1954 and 1973, the times the shown aerial images were acquired.






**Table 2: Attributes of orthophotos used for debris flow mapping. Date of data acquisition for each time step, which covers most of the study area, is marked bold.**

| date of acquisition [y-m-d] | ground resolution per pixel [m] | colours | lower boundary of snow cover [m] |
|---|---|---|---|
| **1947-09-01** | 0.4 | gray-scale | no snow |
| 1947-09-15 | | | |
| 1953-09-08 | 0.25 | gray-scale | 2750 |
| **1954-08-31** | | | |
| 1970-09-10 | 0.2 | gray-scale | 2800 |
| **1973-08-06** | | | |
| 1983-09-24 | 0.5 | gray-scale | 2770 |
| **1990-07-27** | 0.5 | gray-scale | 2400 |
| 1990-10-09 | | | |
| **1997-09-11** | 0.6 | gray-scale | no snow |
| **2003-09-04** | 0.2 | RGB | 2750 |
| **2009-09-08** | 0.2 | RGB | no snow |
| **2010-09-12** | 0.2 | RGB | 2650 |
| **2015-08-03** | 0.2 | RGB | no snow |
| 2017-08-30 | 0.2 | RGB | no snow |
| **2018-09-26** | | | |
| **2020-07-08** | 0.2 | RGB | 2500 |

The mapping and dating of these individual events were carried out in the entire study area and in all available time intervals.

The sub-catchments HT, GT, LT, ZT, WK and FT were considered separately (cf. Fig. 1).

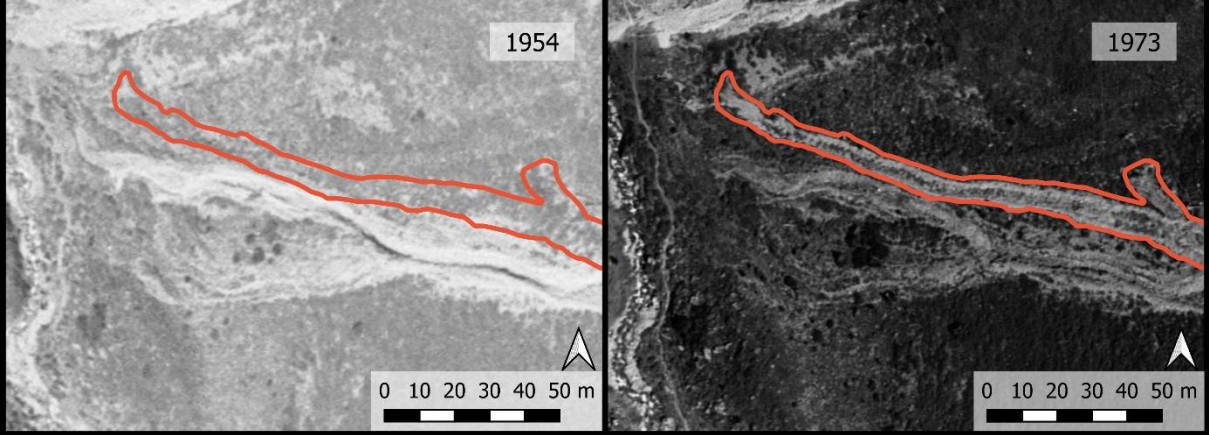

**Figure 3: Example of the mapping process. The mapped debris flow (encircled in red) must have happened between the acquisition of the orthophotos 1954 (no debris flow landforms) and 1973. Sources of aerial images: Province of Tyrol.**



For periods, not all debris flows could be mapped because of poor image quality in shadowed areas. Due to the sometimes
long periods between two orthophotos, especially in the first half of the considered time span, two or more debris flow process
areas might have overlapped in time and space in such way that individual events could not be recorded by the mapping. This
in turn leads to a possible underrepresentation of debris flows, which is more likely in longer time intervals than in shorter
ones.

### 3.2 Volume measurements

We computed the planimetric area of all debris flow deposits except for those where the depositional zone could not be clearly
identified; this was the case with very small events and in shadowed areas.

Because of their high spatial resolution, two different LiDAR datasets were used to determine debris flow deposition volumes
for debris flows which occurred between the single LiDAR epochs. The first dataset from 2006 was available from Land Tirol.
This dataset is only provided as a gridded Digital Terrain Model (DTM; resolution: 1 x 1 m), the initial pointcloud was not
available. The second LiDAR dataset was recorded during a field campaign of the University of Eichstätt-Ingolstadt in 2019
using a Riegl VUX 1LR integrated in a Riegl VP-1 HeliCopterPod (see riegl.com for details). The processing of the raw data
included the precise calculation of the trajectory using the data of two different dGNSS ground stations installed in the study
area. A final strip adjustment was done using the approaches of Glira et al. (2015) and Glira et al. (2016), which are
implemented in the pointcloud processing software OPALS (Pfeifer et al., 2014). The outliers of the resulting pointcloud were
filtered and the ground points were classified using the extension LIS Pro 3D of Laserdata (Petrini-Monteferri et al., 2009) of
the GIS-software SAGA (Conrad et al., 2015). As a result, a final DTM (resolution: 1 x 1 m) could be generated. For more
details about the processing of the raw pointcloud, please refer to Rom et al. (2020).

The difference between the two topographic raster data sets (DoD – DTM of Difference) provided volumetric data for most of
those debris flow depositions that occurred between the LiDAR data acquisitions in 2006 and 2019.

### 3.2.1 Error assessment of the volume data

Uncertainties in the DTMs of 2006 and 2019 lead to errors in the calculated DoD (Lane et al., 2003; Bakker and Lane, 2017)
and thus also in the calculation of the debris flow volumes. In order to minimize the errors in the DoD, the two LiDAR data
sets had to be co-registered. To optimize this processing step, the study area was divided into several smaller areas of interest,
so that the algorithms for matching the data sets were able to work on a more local scale. In these regional patches, areas were
identified where no geomorphologic changes were expected in between the LiDAR data acquisitions. These stable areas were
mapped as close to the debris flow depositions as possible, and were selected, if possible, to be of similar steepness. As
pointcloud data were available only for one of the LiDAR datasets, we coregistered the two gridded DTMs using the approach
by The approach of Nuth and Kääb (2011) implemented in the Python package pybob (https://pybob.readthedocs.io).





To get a better understanding of the errors, the DoD within the identified stable areas were analysed regarding the precision
(standard deviation) and the accuracy (RMSE – root mean square error), as well as the arithmetic mean and the absolute mean.
For the total assessment of the error of the volume of debris flow deposits, the error was calculated following the approach of
Anderson (2019), which combines the uncorrelated random error, the spatially correlated random error and the systematic
error of the DoD. All debris flow volumes detected from the DoD together with the respective errors are listed in the Table in
the Appendix A1.

### 3.2.2 Volume estimation for debris flows not covered by the DoD

The volumes of those debris flow deposits that are detectable in the DoD were determined in each case by summing up the
values of the DoD in the mapped deposition areas. For those debris flow deposits which are not contained in the DoD
(especially for debris flows prior to 2006), only the area of the deposits could be mapped using the respective pair of
orthophotos. In order to estimate the volume of different types of mass movements based on the accumulation area, numerous
studies derived an empirical relationship between the deposit area (A) and the deposit volume (V) (Guzzetti et al., 2009; Magirl
et al., 2010; Larsen et al., 2010). This relationship is expressed by a power-law with an exponent $\gamma > 0$ and the intercept $\alpha$:

$$V = \alpha * A^{\gamma} \tag{1}$$

The exponent $\gamma$ in such area–volume relationships depends not only on the analysed process (e.g. rock fall, landslide, debris
flow), but also on its subtypes (Larsen et al., 2010; Griswold and Iverson, 2008). Nevertheless, the range of $\gamma$ usually seems to
be within a similar range for several types of mass movements (Hilger, 2017).

The relationship between the volumes and deposition areas is used in order to predict the volumes of depositions for which
only the areas are known. Because in this study only slope-type debris flows of the same type are analysed and no large
differences in the debris material are to be expected, the uncertainties here focus on the individual debris flow processes. These
include the different content of water or the topography of the deposition area before the debris flow.

In order to fit Eq. 1 to the empirical data, a variety of different fitting techniques can be used (see for example Guzzetti et al.,
2009; Larsen et al., 2010). One simple method includes a least-squares linear fit to the log-transformed data. Another way of
fitting a power-law function to the data is by using non-linear regression. In order to be better comparable to other studies
calculating such relationships, both methods were applied in the present study by using the statistical software R and the
functions lm() and nls() respectively (Baty et al., 2015).

### 3.2.3 Uncertainties of debris flow volumes

To get a better understanding about the uncertainties involved in the volume calculations, the goodness of fit of the area–
volume models has to be described. However for non-linear correlations, the coefficient of determination $R^2$ is not a valid
measure (Spiess and Neumeyer, 2010). Instead, we use the 95 % prediction interval of the non-linear regressions. The upper
and the lower boundary of the prediction interval for each area was used as the maximum and minimum debris flow volume
for those events that were not quantified from the DoD. Where the lower limit of the prediction interval is negative (which





occurs especially with small deposit areas), it was set to 0 m³; therefore, the lower uncertainty band of the computed volumes is frequently shorter than the upper. For all debris flows included in the DoD, the uncertainty limits were defined by the error assessment in Sect. 3.2.1. For calculating the uncertainty of the total debris flow volume for each of the considered epochs, the uncertainties of each single volume calculation was propagated. Therefore, the total uncertainty of an epoch is the square root of the sum of the squared single uncertainties (Anderson, 2019).

### 3.2.4 Magnitude–frequency relationship

The calculated magnitudes as well as the known ages of the debris flows due to the multi-temporal mapping allowed to establish a magnitude–frequency relationship. This has been done for various gravitational processes, such as landslides (Bennett et al., 2012; Gao et al., 2018; Tanyaş et al., 2019; Guzzetti et al., 2009), rockfalls (Ravanel and Deline, 2011), channelized debris flows (Gao et al., 2018) and also slope-type debris flows (Hilger, 2017). Using the poweRlaw package within R (Gillespie, 2015), we calculated an empirical cumulative distribution function (CDF) to represent the relationship between debris flow deposit volumes and their frequencies (Bennett et al., 2012; Hilger, 2017). Subsequently, we were able to fit a continous power law distribution to the CDF. However, this distribution is only valid for volumes exceeding a minimum magnitude $x_{min}$ (Bennett et al., 2012) and the calculated exponent of the power law β, which is based on a cumulative distribution, has to be reduced by 1 when compared with non-cumulative exponents (Brunetti et al., 2009; Haas et al., 2012).

### 3.3 Hydrological catchment parameters and debris flow magnitudes

The tool "Upslope Area" implemented in SAGA-GIS (Freeman, 1991) was used to calculate the hydrological catchment areas for each of the mapped debris flow starting zones based on the DTM of 2019. With the help of spatial analyses of the catchments we derived a number of different parameters that are known as influencing variables for the magnitude and frequency of debris flows (Wilford et al., 2004; De Haas and Densmore, 2019; Zhao et al., 2020; Zhou et al., 2016). These parameters include the area (A) of the hydrological catchment as well as its length (L), perimeter (P) and mean slope (S). The relief parameter (H) describes the difference between the highest and lowest point of the catchment. The Melton ratio (M) Melton (1957) has been found to correlate with debris flow dynamics (Wilford et al., 2004; De Haas and Densmore, 2019). In addition, relief ratio (R), form factor (F), elongation ratio (E), circularity (C), drainage density (D) and cut density (CD) were calculated according to the definitions in Tab. 3 (see Sect. 4.3). All of the mentioned parameter were correlated to the respective debris flow volumes using Spearman's rho to see a possible connection between the magnitudes and the morphometry of the hydrological catchments.

### 3.4 Analysis of precipitation data

Meteorological data are recorded in the study area at the Horlachalm station, operated by TIWAG at an altitude of 1910 m (see Fig. 1). Temporally high-resolution data (measurements every 15 minutes) are available for precipitation totals since 1989. Since slope-type debris flows in Horlachtal are transport-limited, the frequency of heavy rainfall should be related to the



frequency of debris flow events. Potential triggers are short-term events such as thundestorms rather than days with high total rainfall (Bernard et al., 2020; Underwood et al., 2016; Pelfini and Santilli, 2008). For the Horlachtal, Becht and Rieger (1997) as well as Becht (1995) determined an intensity threshold of 20 mm per 30 min. As the collected data at Horlachalm station only date back to 1989, they cover only a small part of the study period. The meteorological station Längenfeld (provider:
BMNT – Bundesministerium für Nachhaltigkeit und Tourismus) is located further down in the Ötztal valley and has been recording meteorological data since the year 1895. However, these data are not used for evaluation in the present work because the Längenfeld station only records daily totals of precipitation values. Heavy rainfall events of short duration can hardly be reconstructed from daily totals (Pelfini and Santilli, 2008; Jomelli et al., 2007). For example, the statistical evaluation of meteorological data in Altmann et al. (2020) shows that the development of daily totals and heavy rainfall events through
several decades can even be opposite. Nevertheless, daily totals are used in most studies to explain long-term debris flow development (Dietrich and Krautblatter, 2017) because there are hardly any alpine meteorological stations measuring hourly or sub-hourly precipitation totals prior to the 1990s. As temporally high resolution precipitation data are decisive when interpreting long-term debris flow records, it was decided to include the data of the precipitation measuring site Tschagguns (provider: Hydrographischer Dienst Vorarlberg; data available at: https://ehyd.gv.at/). This station records totals for every
minute derived from continuous precipitation data from May 1953 until end of 2018. It is located approximately 80 km west of the study area at an altitude of 681 m (see Fig. 1), but its location north of the Alpine main divide makes the weather conditions comparable to the Horlachtal up to a certain point. Because of the distance between Tschagguns and the Horlachtal, the recorded absolute precipitation data cannot simply be transferred and the extreme precipitation events at Tschagguns are not connected to the debris flow activity in Horlachtal. However, it seems to be promising to analyse trends in high-intensity
precipitation patterns since 1953 to get an idea about changes in extreme event patterns for this part of the eastern Alps.

## 4 Results

### 4.1 Spatio-temporal debris flow mapping

In the entire study area, a total of 834 debris flow events were mapped between 1947 and 2020 using historic and recent orthophotos.
Figure 4 shows the spatial distribution of the mapped process areas. It reveals that the debris flows are not homogeneously distributed over the whole study area, but are mainly concentrated in the three parallel north-south oriented sub-catchments GT, LT and ZT. However, since the sub-catchments vary in size and the periods between the aerial image acquisitions are not uniform, the number of slope-type debris flows per square kilometer and year was calculated for better comparison (Fig. 5).


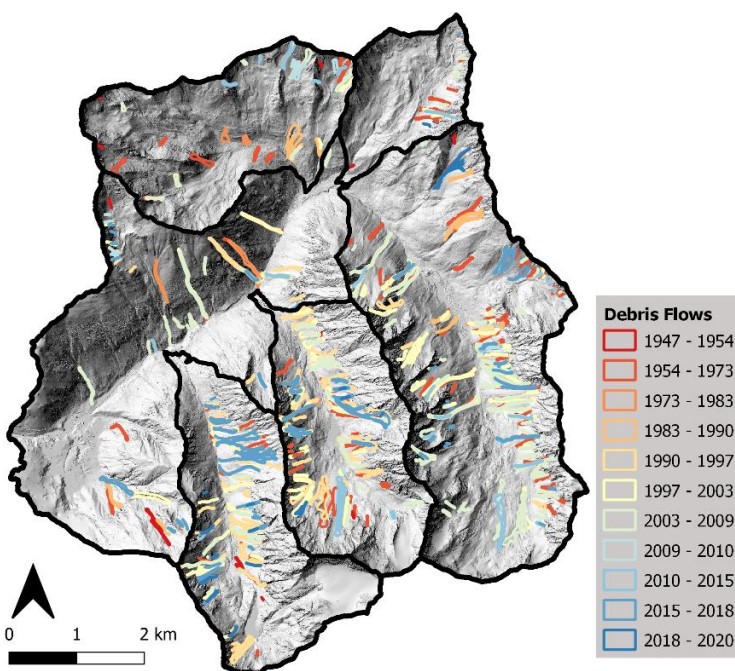

**Figure 4: Results of the debris flow mapping in the entire study area. More recent debris flows overlay older ones in some places.**

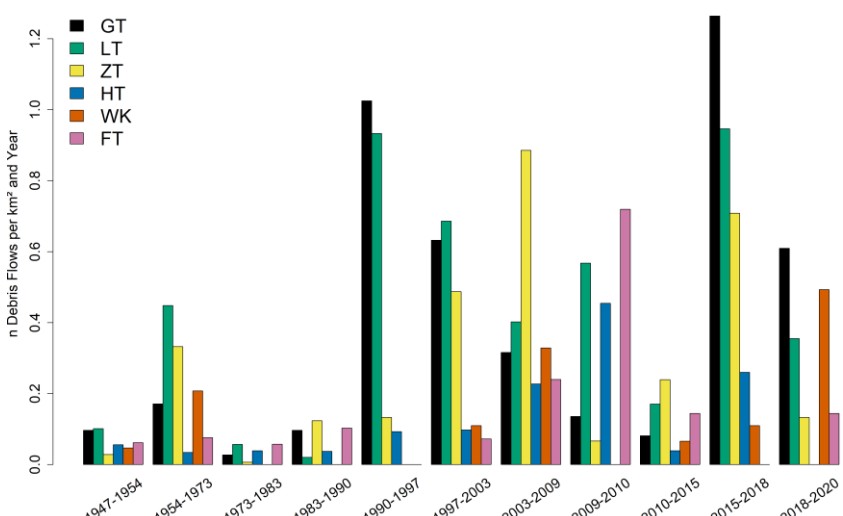

**Figure 5: Mapped debris flows per km² and year. Distinguished between time intervals and sub-catchments.**

The mapped slope-type debris flows show not only spatial but also temporal differences. In Fig. 6, the total number of mapped debris flows in the entire study area (all sub-catchments) for each epoch is depicted. As the time spans of different epochs are not uniform, the annual frequency of debris flows per year is shown as well for a better comparison of the process activity throughout the investigated time span.


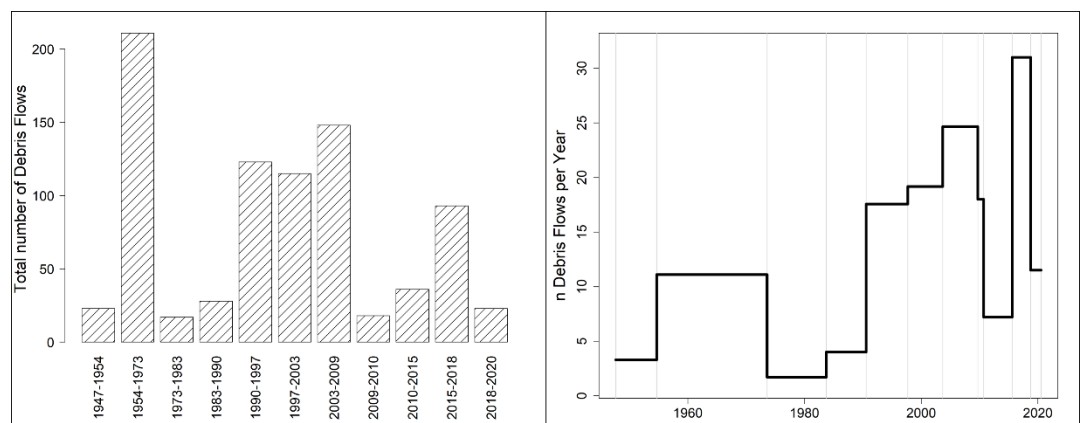

**Figure 6: Temporal variations of slope-type debris flow activity in Horlachtal since 1947. Left: The total number of detected slope-type debris flows in all sub-catchments in each epoch covered by the orthophotos. Right: The number of mapped slope type debris flows per year. The dashed vertical lines represent the acquisition dates of the orthophotos used.**

Periods of higher and lower debris flow activity can be observed in Horlachtal. Between 1954 and 1973, significantly more debris flows were triggered in total and per year than in the periods before (1947–1954) and after (1973–1990). The next very active period lasted from 1990 to 2009. However, the highest number of debris flows per year within the observed time frame occurred between 2015 and 2018.

**4.2 Debris flow volumes**

The mapping of the debris flows showed a concentration of these processes in the parallel sub-catchments GT, LT and ZT. As those debris flows show such a different picture when comparing them to the activity in the other sub-catchments, and because of the similarities in the geomorphological and geographical settings, the analyses concerning deposition volumes were carried out exclusively in GT, LT and ZT.

**Area – volume relationship**

For a total of 58 debris flows, it was possible to map the deposition area in the DoD from the 2006 and 2019 terrain models with sufficient accuracy to enable a balance of the deposition volumes (Appendix A1). The volumes range from very small (7.55 m³) to large debris flows (7506 m³). With the help of this data, a relationship between the area and volume of debris flow deposits could be established which follows a power-law (Fig. 7). The exponent $\gamma$ in Eq. 1 could be calculated as $\gamma = 1.21$ for the fitted linear model. This method tries to fit a linear model to the log-transformed area and log-transformed volume data.

The log scaling of both of the input data results in a distortion of the residuals, which are used in the fitting process.

In order to reduce this bias, a non-linear model was fitted using the nls() function in R. This approach determines the non-linear least-squares estimates of the parameters of a power-law model (Bates and Watts, 1988) and results in a mathematical best fit with respect to the residuals. The exponent of the fitted non-linear model results to be slightly lower with $\gamma = 0.92$ +/- 0.077 for 95 % confidence interval. In the non-linear model plotted in Fig. 7, it is shown that the model slightly overestimates

volumes for areas < 500 m². Again, this is due to the log scaling of the axes.


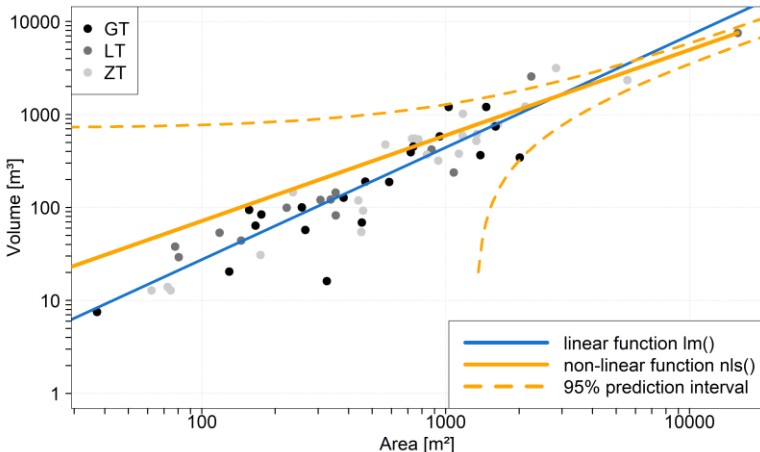

**Figure 7: Relationship between area and volume of debris flow deposits in GT, LT and ZT. The 95 % prediction interval of the model calculated by the non-linear method is shown with the dashed lines.**

With the help of the regression of volume on area, the debris flow volume could be calculated for all events with a precisely

delimited deposit. In total, the deposition volumes could be calculated (based on the DoD) or estimated (area ~ volume) for

404 debris flows in GT, LT and ZT. The uncertainties of the volume calculations were carried out as described in Sect. 3.2.3.

Figure 8 reports both the total and the annual debris flow volume per epoch. Similar to the mapping results, the debris flow

volumes show periods with high and low deposition rates per year.

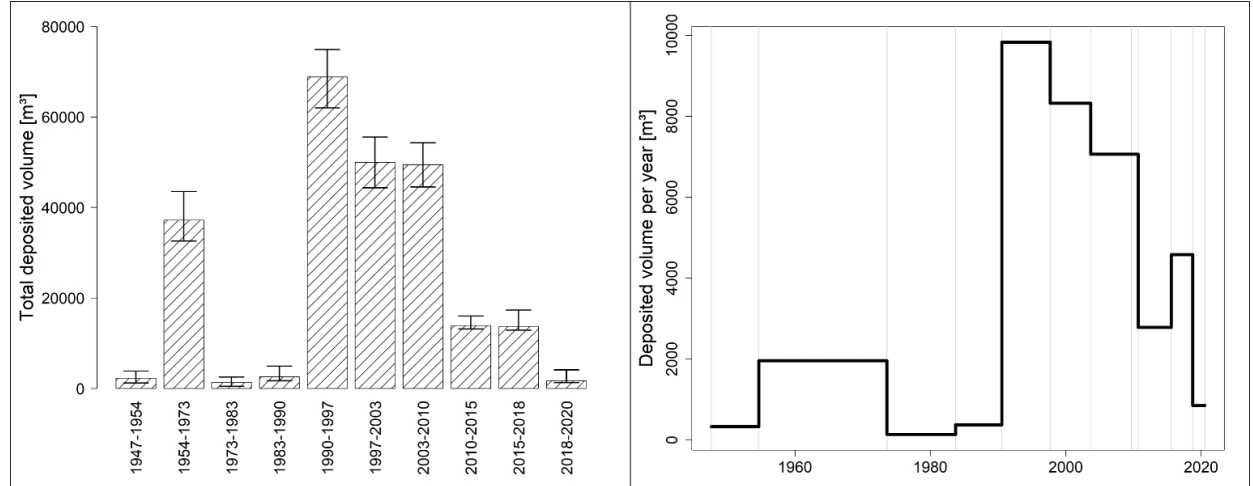

**Figure 8: Temporal variability of the volumes deposited by slope-type debris flows in Horlachtal since 1947. Left: The total deposited**
295 **volume of slope-type debris flows in all sub-catchments in each epoch covered by the orthophotos. Uncertainties of the calculations**
**are added for each time span. Right: The deposited volume of slope type debris flows per year. The dashed vertical lines represent**
**the acquisition dates of the orthophotos used.**

Again, the results of the time intervals between 1954–1973 and 1990–2010 reveal increased deposition, but the period between

2015 and 2018 produced less deposited volume than one could have assumed from the very high number of triggered debris





flows in that time interval. Thus, it can be stated that although manyevents occurred, they have deposited relatively little

sediment in total.

**Magnitude–frequency relationship**

The calculated magnitude–frequency relationship is shown in Fig. 9. About 70 % of all debris flows in GT, LT and ZT have

an accumulated volume above 100 m³ and about 20 % of all debris flows exceed a volume of 1000 m³. Extreme events, which

account for less than 1 % of all debris flows, can reach volumes of more than 10000 m³. The magnitude–frequency relationship

can be described by a continous power-law distribution with an exponent of β = 2.9 for volumes of $x_{min}$ = 1025 m³ and above

and therefore is especially usable for large debris flow volumes.

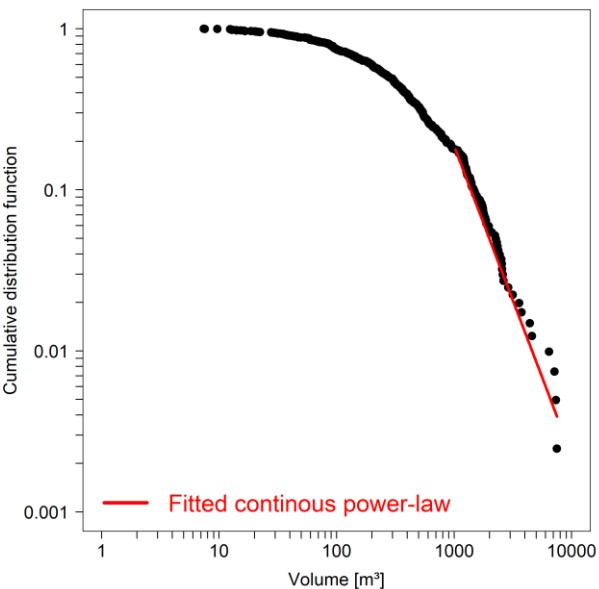

**Figure 9: Magnitude–frequency relationship displayed with a cumulative distribution function for the debris flows in GT, LT and**
**ZT. The fitted continous power-law describes the relationship for debris flows with a volume greater than 1025 m³.**

**4.3 Analysis of hydrological catchment parameters**

For all 404 debris flows for which the volume could be determined, we performed a correlation analysis of the volumes with

various parameters of the respective hydrological catchment areas (Tab. 3).

Although no variable shows a very strong (positively or negatively) correlation, Spearman's rho point to slightly positive

interrelationships between debris flow volumes and A, L, P, H, F as well as E. In addition, D and CD indicate a negative

correlation in the same order of magnitude. Especially the variables S and C on the other hand have no visible influence on the

debris flow volumes. However, this analyses show that the morphometry of the hydrological catchments indeed have influence

on the slope-type debris flow magnitudes.

**Table 3: Calculated parameters of the hydrological catchments of the slope-type debris flows alongside their definitions and**
**dimensions. Correlation of each parameter with the respective debris flow volumes were calculated by Spearman's rho. The p-values**
**represent the significance of the correlations.s**

| catchment parameter | index and definition | dimension | Spearman's rho | p-value |
|---|---|---|---|---|
| planimetric area | A | m² | 0.46 | < 2.2e-16 |
| length | L | m | 0.43 | < 2.2e-16 |
| perimeter | P | m | 0.46 | < 2.2e-16 |
| mean slope | S | degree | 0.02 | 0.74 |
| relief | H | m | 0.43 | < 2.2e-16 |
| Melton ratio | $M=H/\sqrt{A}$ | - | -0.22 | 1.1e-5 |
| relief ratio | $R=H/L$ | - | -0.11 | 0.02 |
| form factor | $F=(A/L)^2$ | m | 0.45 | < 2.2e-16 |
| elongation ratio | $E=(4*A/\pi)/L$ | m | 0.45 | < 2.2e-16 |
| circularity | $C=(4*\pi*A)/P^2$ | - | 0.05 | 0.29 |
| drainage density | $D=L/A$ | m$^{-1}$ | -0.45 | < 2.2e-16 |
| cut density | $CD=R/P$ | - | -0.43 | < 2.2e-16 |

## 4.4 Precipitation analysis

For the initiation of debris flows in Horlachtal, high intensity rainfall events with a large amount of precipitation within a short
time are necessary. Therefore, the precipitation data of the stations Horlachalm and Tschagguns were analysed regarding high
intensity events. In Fig. 10, the recorded rainfall data are shown in mm per 30 min for both stations. In order to determine a
possible increasing or decreasing trend in rainfall events reaching high intensities within a short time interval, all days on
which 10 mm per 30 min was exceeded were marked in the records of both stations. The exact magnitude of this threshold is
not of great importance because it is only used in order to get an idea about possible trends in high intensity rainfall events. As
the threshold of 20 mm per 30 min turned out be exceeded quite rarely in both rainfall records, it was set to 10 mm per 30 min.
However, the frequency of these extreme events do not show statistically significant increases or decreases as can be seen in
Fig. 10.

The cumulative sums of days with a high intensity rainfall event exceeding 10 mm per 30 min for both stations are shown in
Fig. 11. On average, there are slightly more such events per year recorded at Tschagguns station (1.093) than at Horlachalm
station (0.831). Nevertheless, there are hardly any periods with particularly many or few events that stand out. The longest
time without event at Tschagguns is nearly five years between 28 August 1995 and 14 June 2000 and at Horlachalm almost
four years between 05 August 2004 and 26 June 2008. But temporal trends of more or few events per year are detectable
neither at Tschagguns nor at Horlachalm.


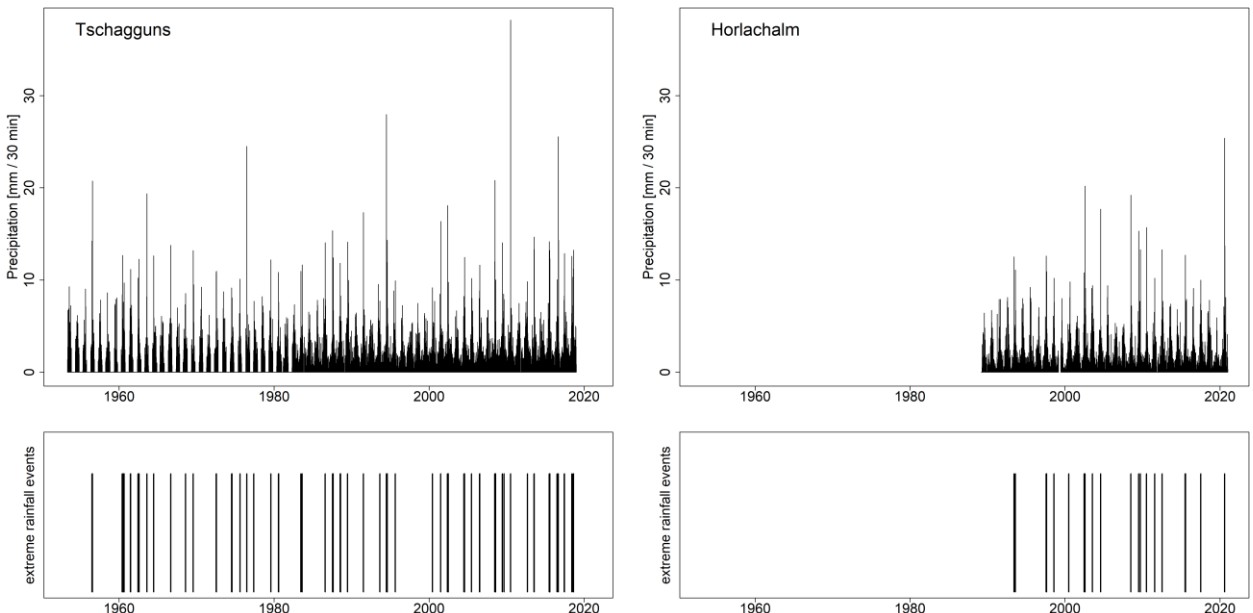

**Figure 10: Recorded precipitation data of the meteorological stations Tschagguns (left) and Horlachalm (right). Days with intensities**
**exceeding 10 mm per 30 minutes are marked in the respective bottom plots.**

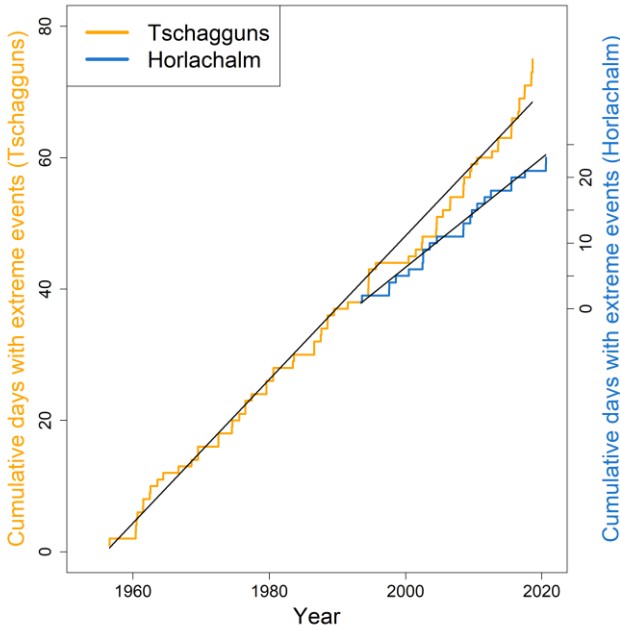

**Figure 11: Cumulative sums of days with precipitation intensities exceeding 10 mm per 30 minutes for Tschagguns (orange) and Horlachalm (blue).**





## 5 Discussion

### 5.1 Spatial variability of slope-type debris flows

**Methodological limitations**

The catchment-wide mapping of debris flows using historical and recent orthophotos showed spatial differences within the study area (see Fig. 4). Most of the debris flow events were mapped in the three parallel north-south oriented sub-catchments GT, LT and ZT. The reason for this pattern cannot be clarified conclusively and might be the focus of further investigations.

Compared to GT, LT and ZT, the debris flow accumulation zones of the mainstem valley HT are located at lower altitudes and are covered by high vegetation, especially forested areas. In such regions it is much harder to identify new debris flow accumulations in orthophotos, especially if the quality of the orthophotos is not good and the debris flow magnitude is low. This could result in an underestimation of mapped debris flow events in HT.

**Topographic variability**

The biggest factor for the concentration of debris flow processes is probably the presence of catchments in the steep bedrock above large talus cones, which is typical for slope-type debris flows (Rieger, 1999). Rainwater concentrates at the contact zone between these catchment areas and the slope sediments underneath and potentially triggers debris flows. These catchment morphometries are especially pronounced in GT, LT and ZT.

In the very active and north-south oriented sub-valleys GT, LT and ZT, 68 % of the active debris flow catchments face west,

while only 32 % face east. Becht (1995) attributes this differences to the emergence of cirques in the Pleistocene on east-exposed slopes. West-exposed ones do not show these landforms. Therefore, the stepped profiles on east exposed slopes caused by the cirques prevent the accumulation of high peak discharges during a rainfall event because of a buffering effect of these cirques. Depending on the amount of loose material in the cirques, these buffering effects can cause longer or shorter delays of the runoff and therefore lower the peak discharge. In addition, the slopes beneath the cirques lack rockfall material supply,

which otherwise works as suitable material on the slopes for debris flow initiation.

**Rainfall variability**

It is noticeable that in certain time intervals some sub-catchments were significantly more affected by debris flows than other sub-catchments (see Fig. 5). This is most obvious in the period between 1990 and 1997, when the two neighbouring valleys GT and LT show a strongly increased debris flow activity, especially in comparison to the other sub-catchments. Other time

intervals (2003–2009, 2009–2010) also show that some sub-catchments were obviously more affected than neighbouring ones. We attribute these findings to the fact that debris flow-triggering heavy rainfall events often occur during intense convective and thus spatially restricted thunderstorms (Underwood et al., 2016; Berti et al., 2020; Stoffel et al., 2005) that often affect only parts of the study area.

**Effect of debris flow catchment parameter**

The results of the correlation of the debris flow volumes and the parameters of the respective hydrological catchments in Tab. 3 indicate that the catchment morphometry is affecting the spatial variability of slope-type debris flows in Horlachtal. Only a



few debris flow studies implemented analyses regarding catchment parameters and most of them deal with different debris flow types and scales than this study (Becht and Rieger, 1997; Wilford et al., 2004; Marchi et al., 2019; Li et al., 2015). However, the results of this study match the findings of De Haas and Densmore (2019), who worked in a roughly comparable

setting in the United States and found statistically significant correlations between debris flow lobe volumes and A, L, P, H and M. This fits our data just as well as the lack of correlation with S, C and R. The only differences are with the variables F and E, which show a stronger relationship in Horlachtal compared to the results in De Haas and Densmore (2019). In addition, a positive correlation between slope-type debris flow volumes and A was detected by Rieger (1999) in LT as well.

**5.2 Temporal variability of slope-type debris flows**

**5.2.1 Frequencies and magnitudes in different periods**

**Methodological limitations**

Geomorphological mapping using historic and recent orthophotos is a suitable tool to generate debris flow records for larger study areas, as presented here. A big advantage is that the aerial images cover the entire study area in great detail. It is therefore possible to generate a near complete debris flow record. Problems only occur when the quality (in terms of resolution, shadow

effects and snow cover) of the used images is poor, or more than one debris flows occur during one epoch and the process areas overlap. Disadvantages of this method however are the sometimes large time spans between the acquisition dates of two consecutive aerial images. This not only leads to an increased probability of overlapping events, but the dating of single debris flows becomes quite inaccurate. In addition, the durations of different epochs are not equal. Therefore, in order to compare the debris flow activity, the normalised number of debris flows per year was calculated. It has to be mentioned, however, that

slope-type debris flows are triggered by single precipitation events. The calculations of "debris flows per year" suggest a uniformly distributed debris flows activity throughout the respective epochs, which is far from reality, and hence these calculations should be treated with caution.

Because of the different durations of the epochs, the debris flow record is slightly biased, as the number of debris flows in longer intervals (e.g. 1954–1973) is likely to be underestimated because of the aforementioned reasons. This can be

investigated by deliberately removing orthophotos, mapping events based on the remaining imagery, and comparing the resulting map with the complete record. In order to get an idea about the magnitude of debris flow underestimation in longer epochs, a re-mapping of debris flow processes was done in GT and LT, where the orthophotos of 1997 and 2003 as well as 2010, 2015 and 2018 were removed from the record. The number of detected debris flows of this second test mapping was then compared with the number of detected debris flows in the original record (reference) in the same timeframe (Tab. 4).

Compared to the reference data, between 22 % and 29 % of the debris flows were missed in the test mapping because of the missing orthophotos in between the timeframes. These results indicate that in longer epochs (e.g. 1954 – 1973) the number of slope-type debris flows are underestimated by about 25 % in relation to shorter epochs (e.g. 2015 – 2018) due to overlapping process areas for example.


**Table 4: Comparison of the number of detected slope-type debris flows of the test mapping with a reduced number of orthophotos and the original mapping (reference). This mapping was conducted in GT and LT and for the time slices 1990 – 2009 and 2009 – 2020.**

| sub-catchment | 1990 – 2009 Reference | 1990 – 2009 Test | 2009 – 2020 Reference | 2009 – 2020 Test |
|---|---|---|---|---|
| GT | 95 | 74 | 41 | 29 |
| LT | 92 | 67 | 35 | 27 |

**Relationship between frequency and magnitude**

With the help of the area ~ volume relationship we were able to calculate many deposit volumes, which is necessary to establish a relationship between frequency and magnitude. The resulting exponent of γ = 0.92 of the nls() method connecting volumes and areas is comparable with the result of Hilger (2017), who used a linear regression method for slope-type debris flows in Kaunertal (γ = 1.08). Furthermore, it is consistent with other comparable studies (Jaboyedoff et al., 2020; Larsen et al., 2010; Guzzetti et al., 2009). The established relationship enabled detailed frequency – magnitude analyses. There are rarely any other studies that calculate such a magnitude–frequency relationship only for debris flows and especially slope-type debris flows. Other studies often focus on debris flows in general (e.g. Hungr et al., 2008; Riley et al., 2013). An exception is Hilger (2017), who performed such a calculation in a similar geological setting using slope-type debris flows at Kaunertal, Tyrol. While the general shape and β are very much comparable to this study, the magnitudes of debris flows in Horlachtal are larger by an order of a magnitude. Thus, the largest debris flows in Kaunertal show a volume of about 1000 m³, whereas in Horlachtal the volumes can reach nearly 10000 m³ (Fig. 12 (a)). In addition, a slight shift of debris flow magnitudes could be detected in Horlachtal. The debris flows of the second half of the investigated time span (1983 – 2020) reach very high volumes more often than debris flows of the first half (1947 – 1983) (Fig. 12 (a)).

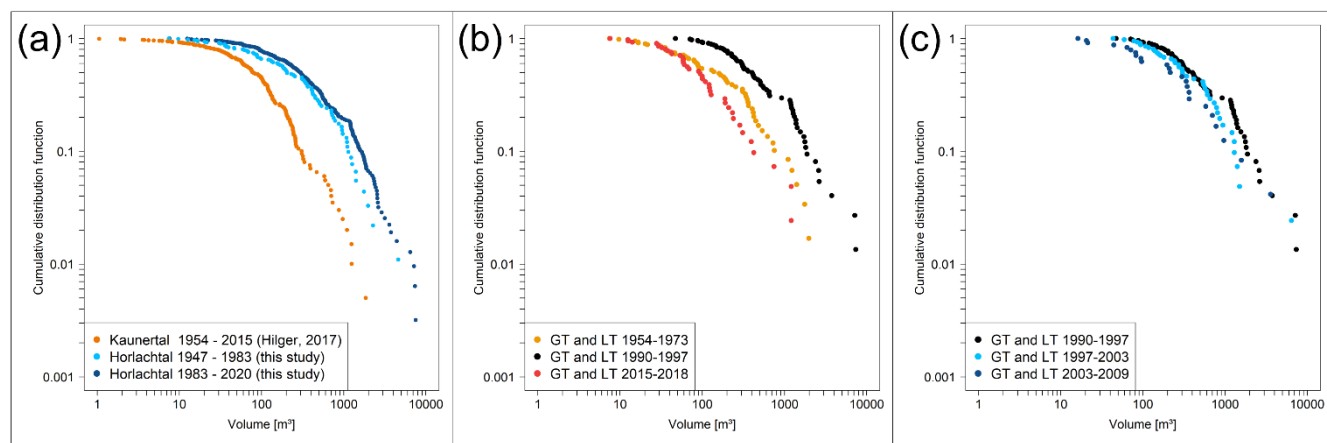

**Figure 12: (a): Magnitude – frequency relationship for the early debris flows in Horlachtal (1947 – 1983, light blue) compared with recent debris flows (1983 – 2020, dark blue). The results of a similar investigation from Hilger (2017) in Kaunertal is shown in orange. (b): Comparison of the magnitude – frequency relationships of the three highly active debris flow periods in GT and LT. (c): Comparison of the magnitude – frequency relationships for the three consecutive periods 1990–1997, 1997–2003 and 2003–2009 in GT and LT.**





**Temporal development of slope-type debris flow activity**

In Horlachtal, the multi-temporal mapping of debris flows as well as the volumetric measurements resulted in periods with higher activity (1954–1973, 1990–2010 and 2015–2018) and lower activity in between (Fig. 6). A consistent linear trend is
not recognizable although it seems that in periods with low activity, the number of debris flows has been rising since 1947. This finding might be biased because of two limitations. First, the most recent orthophotos (especially since 2003) are of very good quality, especially in terms of spatial resolution, light conditions and extent of snow cover. These circumstances allow even small events to be detected and mapped. The second reason is the aforementioned difference in time span lengths, which results in the underestimation of detected debris flows in longer periods.

The mapping results as well as the calculated deposition volumes indicate no long-term change in debris flow activity in the past 70 years but with some short-term variabilities. This matches other debris flow records in the Alps (Bollschweiler and Stoffel, 2010; Bollschweiler et al., 2008; Jomelli et al., 2007). Kiefer et al. (2021) detected significant changes over substantially longer time periods when reconstructing the debris flow activity for the past 4000 years based on turbidite measurements of a fan delta. But even in this record, no significant change within the last century is recognizable.

In Horlachtal, three periods of increased slope-type debris flow activity can be recognized based on the total number of detected processes (see Fig. 6): Between 1954 and 1973, from 1990 to 2009 and from 2015 to 2018. A particularly large debris flow event was reported for 31 July 1992, mainly affecting GT and LT (Becht, 1995). These statements can be supported by the debris flow mapping of this study, as many events could be detected in the period 1990–1997 especially in GT and LT (see Fig. 5). In the two subsequent periods until 2009, still many events can be registered in GT and LT, but with a decreasing
tendency. This could indicate that the large debris flow event in 1992 still had some kind of impact on the debris flow activity of the following years due to the disturbance of the system like e.g. destruction of vegetation or channel deepening. Such a kind of impact on subsequent periods could only be detected for the 1992 event. The high debris flow activities between 1954–1973 and 2015–2018 did not show this pattern of aftereffects, at least not for the temporal resolution predetermined by the orthophotos.

**Debris flow magnitude comparison of highly active periods**

An indication of the outstanding significance of the 1992 event in GT and LT compared to 1954–1973 and 2015–2018 can be provided by considering the debris flow magnitudes. Figure 12 (b) shows that the deposited debris flow volumes of the period 1990–1997 (which includes the 1992 event) are significantly higher in comparison with the magnitudes of 1954–1973 and 2015–2018, respectively. In addition, Fig. 12 (c) shows a slightly decreasing tendency in deposited debris flow volumes from
1990–2009, which in turn supports the aforementioned impact of the 1992 event on the following years.

In the heatmap of Fig. 13, each row represents one slope-type debris flow starting zone in GT or LT, which was active at least in two different periods between 1947 and 2020. For each individual starting zone (each row of the heatmap), the magnitudes were normalized between 1 (largest event of the starting zone; dark colouring) and 0 (smallest event of the starting zone; light colouring). If no debris flow could be detected at a starting zone in a specific period, the respective colour was set to grey. The
heatmap shows that out of the 82 different starting zones in GT and LT which were active at least twice, the largest debris flow





event could be detected within the period 1990–1997 on 42 occasions. 73 % of the starting zones produced their maximum deposited volume between 1990 and 2009, however showing a decreasing tendency. Only on 11 different occasions, the largest debris flows could be detected in the period 1954–1973 and within 2015–2018, only four starting zones produced their largest event of the total considered timeframe.

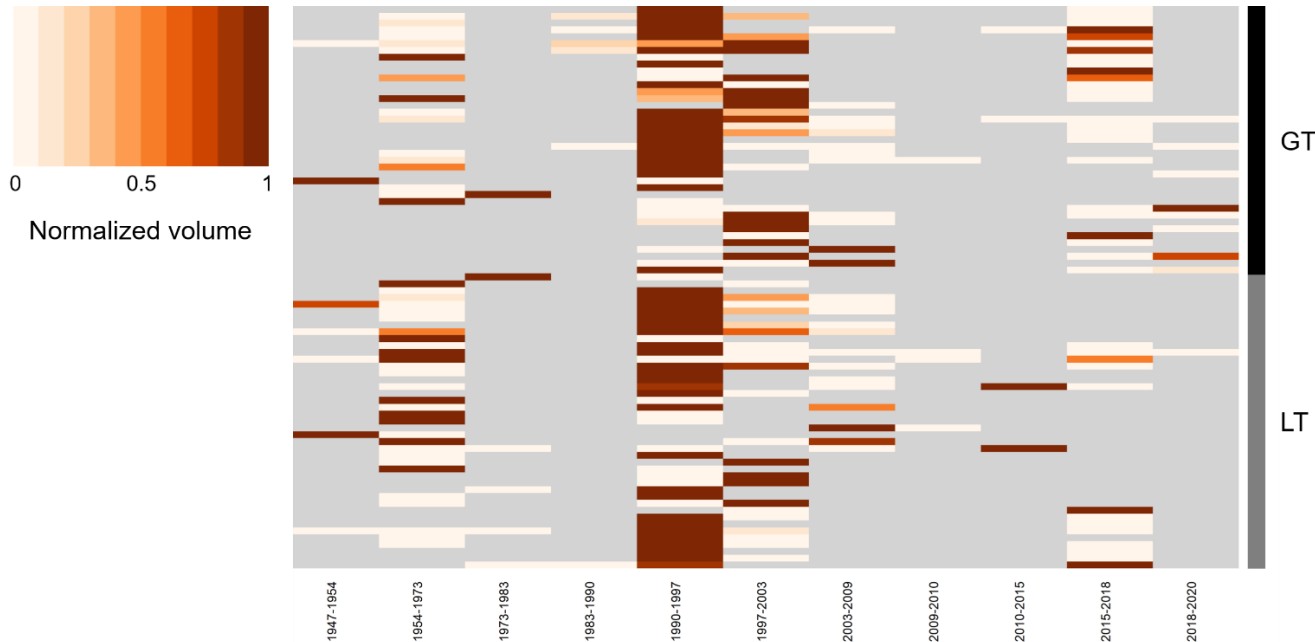


**Figure 13: Heatmap of the 82 different starting zones in GT and LT which were triggered at least twice between 1947 and 2020. The normalized magnitudes of every starting zone is showed by the colouring scheme. For more details please refer to the text.**

The results of the magnitude comparisons of the most active periods indicate a strong influence of the 1990–1997 event as it produced the largest debris flow volumes in GT and LT by far. This in turn supports the assumption that the debris flows of

1992 (Becht, 1995) affected the debris flow activity in GT and LT for the following years and the system needed some time to reach the state of before 1990.

The highly active period 1990–2009 with the highest debris flow magnitudes might have affected the debris flow system for even longer. The discrepancy between the high number of detected debris flows from 2015 to 2018 and the relatively small deposited volumes in the same period possibly points to recharge time effects of debris flow channels, as mentioned in Pelfini

and Santilli (2008) and demonstrated in Jakob et al. (2005), Jakob et al. (2020) and Berger et al. (2011). During the highly active period (1990–2009) the sediment storages were depleted so that the debris flows triggered afterwards showed below-average magnitudes. This in turn indicates temporary material limitation of the slope-type debris flow systems in Horlachtal.


### 5.2.2 Precipitation and debris flow activity in Horlachtal

Rainfall events triggering slope-type debris flows can occur very locally (see Sect. 5.1). The precipitation data from the
Horlachalm meteorological station can therefore not be related to debris flow processes in the entire Horlachtal. As a
consequence, the calculation of triggering thresholds using intensity-duration relationships or other methods (Berti et al., 2020;
Segoni et al., 2018) would thus be rather inaccurate. Due to the location of the meteorological station, the data are only set in
relation to debris flows in ZT. The threshold value for debris flow triggering of 20 mm / 30 min according to Becht and Rieger
(1997) has only been exceeded on two days since 1989, namely on 31 July 2002 and 02 July 2009. Nevertheless, the mapping
of debris flow processes in ZT since 1947 shows that debris flows have also occurred during periods in which the threshold
value of 20 mm / 30 min was not exceeded at the Horlachalm meteorological station. This in turn indicates that a precipitation
event < 20 mm / 30 min in the study area can be sufficient to trigger debris flows.

In Fig. 14, all high intensity rainfall events recorded at the Horlachalm station with precipitation values exceeding 5 mm per
30 minutes are shown together with the mapped number of debris flows per year in ZT and the calculated deposition volume
per year in ZT. The acquisition date of the orthophotos, which were used for mapping the debris flows are marked as grey
vertical lines the Figure.

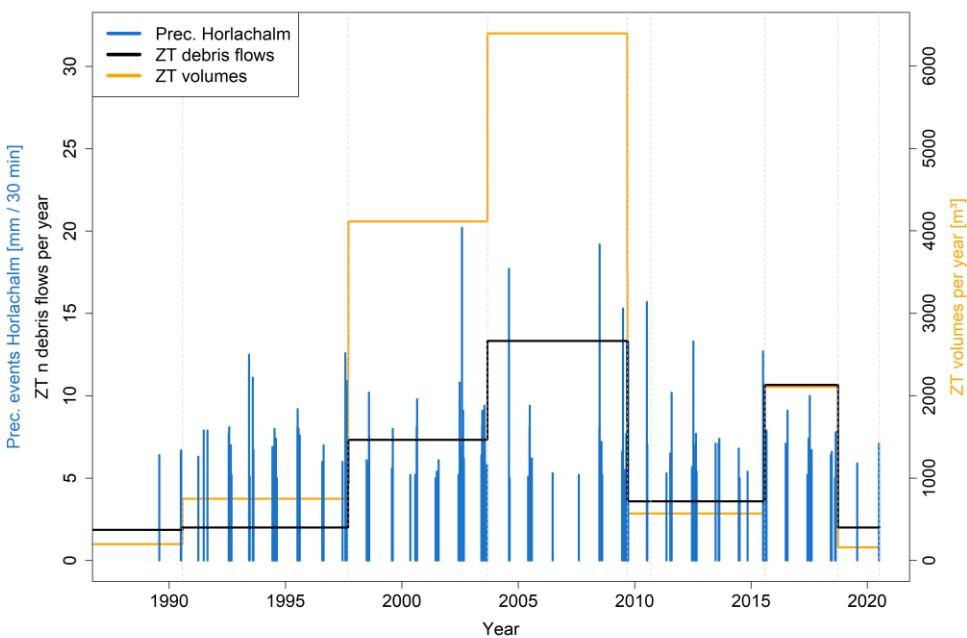

**Figure 14: High intensity rainfall events exceeding 5 mm per 30 minutes at the Horlachalm station combined with the number of mapped debris flows per year in ZT (black) and the deposited debris flow volumes per year in ZT (orange). The sub-catchment ZT**
**is located quite close to the Horlachalm meteorological station.**





Only very few debris flows were mapped in the period between 1990 and 1997 in ZT. The precipitation data show that between the times of the aerial images acquisition in 1990 and 1997, precipitation events of over 10 mm / 30 min were indeed recorded only rarely, and these few precipitation peaks always remained below 15 mm / 30 min. During that period, the precipitation value of 10 mm / 30 min was exceeded a total of 5 times on four different days. The maximum value is 12.6 mm / 30 min on
28 July 1997.

Between the aerial images acquisition in 1997 and 2003, however, considerably more numbers of debris flows and higher debris flow volumes per year were detected in ZT. The precipitation records also show more extreme rainfall events during this period. Precipitation exceeded the value of 10 mm per 30 minutes a total of 11 times on six different days. The maximum values are also far above the level of the last period, with a maximum value of 20.2 mm / 30 min on 31 July 2002. Even more
debris flow processes per year were mapped in the period between 2003 and 2009.

On four days between the acquisition of the aerial images of 2010 and 2015, the value of 10 mm / 30 min was exceeded six times. However, the maximum of 13.9 mm / 30 min on 07 July 2015 is again relatively low. During this period, comparatively few debris flows per year were mapped in ZT. In the most recent time step between 2015 and 2018, the threshold value was only reached once at the Horlachalm meteorological station on 10 July 2017. At exactly 10.0 mm / 30 min, the maximum value
for this period is relatively low. However, many debris flows per year could be determined in ZT during this time step. The corresponding debris flow triggering event cannot be traced in the precipitation data. It is therefore probable that this must have been a very local rainfall event that strongly affected ZT, but could only be measured at a lower level at the Horlachalm meteorological station.

In general, a correlation between debris flow activity in ZT and precipitation data at Horlachalm station is recognisable. The
highest precipitation intensities per 30 minutes were recorded in the time steps 1997–2003 and 2003–2009, and also many debris flow processes could be mapped during this epoch in ZT. During the periods 1990–1997 and 2010–2015 the maximum precipitation per 30 minutes was significantly lower, which is also reflected in the lower number of mapped debris flows per year. Only in the time steps 2015–2018 the data sets do not seem to match. This contrast is interpreted as a further indication for very local rainfall events as triggers of debris flows.

The evaluation of high intensity rainfall events in combination with the mapped slope-type debris flows has shown that the threshold of 20 mm / 30 min specified by Becht and Rieger (1997) is indeed sufficient to trigger large debris flow events in the study area. However, depending on the preconditions, even lower precipitation intensities seem to be sufficient to start debris flows on very active debris cones.

## 6 Conclusion

**Methods and data**

In this study, a long-term development of slope-type debris flows within a large alpine catchment is presented. With the help of adetailed mapping using area-wide orthophotos, 834 different slope-type debris flows have been identified to have occurred




between 1947 and 2020 in the Horlachtal. The advantage of establishing a debris flow record of an alpine catchment based on orthophotos is the area-wide high spatial resolution of each time step. Problems occur, however, because of the sometimes

long time interval between the acquisitions of two consecutive orthophotos. Test mappings using a different set of orthophotos showed that in long time steps the number of debris flows is underestimated by about 25 % compared to shorter time steps. Another difficulty of the used method is the inaccurate dating of debris flow events. As a result, temporal changes in debris flow activity can only be detected by normalizing the mapped debris flows for each period, which contradicts the sudden and rapid nature of these processes.

Integrating two spatial high resolution LiDAR datasets enabled the establishment of a relationship between deposition area and deposited volume. Using this relationship, it was possible to calculate deposited slope-type debris flow volumes even for past debris flow processes.

This large spatio-temporal records of mapped debris flows and corresponding volumes revealed spatial as well as temporal variabilities in the debris flow dynamics in the study area.

**Spatial variability**

High slope-type debris flow activity in Horlachtal is concentrating on the west exposed slopes of the three parallel sub-catchments GT, LT and ZT. The detailed mapping of debris flows revealed that single debris flow events can occur very locally and thus point to the very local characteristics of debris flow triggering rainfall events. This in turn reveals that the data of one meteorological station within an alpine catchment cannot explain the entirety of the debris flow dynamics for that region.

A very important factor for the occurrence of slope-type debris flow is a suitable hydrological catchment in bedrock above the slope sediments. Analyses of the hydrological catchments of the debris flows in GT, LT and ZT showed that their attributes like area, length, perimeter and relief ratio have substantial influence on the initiation and magnitude of slope-type debris flows and might be subjects for further investigations.

**Temporal variability**

A consistent trend in slope-type debris flow activity in Horlachtal since 1947 cannot be detected. However, periods with enhanced debris flow activity (1954–1973, 1990–2009, 2015–2018) and periods with lower activity alternate with one another. Based on comparisons in frequencies and magnitudes as well as a heatmap analysis of the debris flow volumes, it can be stated that the debris flow event between 1990 and 1997 in GT and LT not only showed the largest deposited volumes but had an impact on the debris flow system for the following years. Between 1997 and 2009 the debris flow activity in GT and LT was

still high, but decreased in frequencies and magnitudes. This highly active phase might have even influenced the debris flow event 2015–2018. The clearly below average volumes of the latter might be caused by recharge time effects of the sediment storages, which were emptied during 1990–2009.

The most important factor for triggering debris flows in Horlachtal are extreme rainfall events with high intensities within a short timeframe. Long-term data of alpine precipitation records with a sub-hourly resolution are very rare but the measurements

of the Horlachalm meteorological station since 1989 as well as the Tschagguns station since 1953 do not show any statistical significant increases or decreases of days with an extreme rainfall event.





**Outlook**

For a better understanding of long-term slope-type debris flow behavior, more studies in alpine catchments with long investigations periods are necessary. Such long-term investigations are difficult because debris flows on slopes usually do not
affect infrastructure and thus are not recorded in written archives. Since slope-type debris flows, unlike channelized debris flows, are not as affected by anthropogenic interventions, they are very important to investigate the influence of climate changes to the geomorphodynamics of high alpine landscapes. In order to overcome the mentioned difficulties of debris flow records based on orthophotos, combining it with more methods (Jomelli et al., 2007) like dendrogeomorphology, lichenometry or the evaluation of historical documents might be useful. In addition, an expansion of the debris flow record in Horlachtal
prior to 1947 might provide an even better understanding of slope-type debris flow dynamics over longer timespans and may help to understand the role of the forcing during different climate conditions.



## Appendix

**Table A1: Error assessment of debris flow deposit volumes between 2006 and 2019 following Anderson (2019).**

| Debris flow ID | Volume [m³] | Standard deviation [m] | RMSE [m] | Arithmetic mean [m] | Absolute mean [m] | Uncorrelated random error [m³] | Spatially correlated random error [m³] | Systematic error [m³] | Total error [m³] |
|---|---|---|---|---|---|---|---|---|---|
| 123 | 63.899 | 0.124 | 0.124 | 0.007 | 0.088 | 1.612 | 1.899 | 1.126 | 2.734 |
| 124 | 365.867 | 0.124 | 0.124 | 0.007 | 0.088 | 4.625 | 5.446 | 9.259 | 11.695 |
| 127 | 16.218 | 0.124 | 0.124 | 0.007 | 0.088 | 2.243 | 2.641 | 2.178 | 4.092 |
| 130 | 20.473 | 0.113 | 0.113 | 0.002 | 0.086 | 1.285 | 0.000 | 0.221 | 1.304 |
| 131 | 344.987 | 0.113 | 0.113 | 0.002 | 0.086 | 5.054 | 0.000 | 3.419 | 6.102 |
| 132 | 582.765 | 0.113 | 0.113 | 0.002 | 0.086 | 3.470 | 0.000 | 1.612 | 3.826 |
| 136 | 84.162 | 0.113 | 0.113 | 0.002 | 0.086 | 1.495 | 0.000 | 0.299 | 1.525 |
| 140 | 94.188 | 0.226 | 0.226 | 0.007 | 0.085 | 2.806 | 0.000 | 1.016 | 2.984 |
| 141 | 1208.687 | 0.226 | 0.226 | 0.007 | 0.085 | 8.639 | 0.000 | 9.636 | 12.942 |
| 142 | 743.536 | 0.226 | 0.226 | 0.007 | 0.085 | 9.030 | 0.000 | 10.527 | 13.869 |
| 144 | 1207.066 | 0.226 | 0.226 | 0.007 | 0.085 | 7.242 | 0.000 | 6.772 | 9.915 |
| 145 | 190.190 | 0.226 | 0.226 | 0.007 | 0.085 | 4.881 | 0.000 | 3.076 | 5.769 |
| 148 | 393.834 | 0.226 | 0.226 | 0.007 | 0.085 | 6.054 | 0.000 | 4.732 | 7.684 |
| 149 | 127.653 | 0.226 | 0.226 | 0.007 | 0.085 | 4.419 | 0.000 | 2.521 | 5.088 |
| 151 | 188.315 | 0.226 | 0.226 | 0.007 | 0.085 | 5.459 | 0.000 | 3.848 | 6.679 |
| 160 | 57.467 | 0.113 | 0.113 | 0.002 | 0.086 | 1.835 | 0.000 | 0.451 | 1.889 |
| 160 | 82.222 | 0.126 | 0.126 | 0.003 | 0.083 | 2.371 | 0.634 | 1.059 | 2.673 |
| 161 | 69.029 | 0.113 | 0.113 | 0.002 | 0.086 | 2.391 | 0.000 | 0.765 | 2.510 |
| 163 | 7.548 | 0.113 | 0.113 | 0.002 | 0.086 | 0.676 | 0.000 | 0.061 | 0.679 |
| 164 | 100.305 | 0.113 | 0.113 | 0.002 | 0.086 | 1.803 | 0.000 | 0.435 | 1.855 |
| 165 | 2562.894 | 0.102 | 0.102 | 0.001 | 0.077 | 4.656 | 0.000 | 2.918 | 5.495 |
| 168 | 7506.207 | 0.126 | 0.126 | 0.003 | 0.083 | 15.837 | 4.234 | 47.244 | 50.007 |
| 169 | 452.303 | 0.170 | 0.171 | -0.008 | 0.116 | 4.631 | 0.000 | -6.117 | 7.673 |
| 171 | 53.523 | 0.101 | 0.101 | 0.000 | 0.077 | 1.101 | 0.000 | 0.012 | 1.101 |
| 171 | 120.910 | 0.101 | 0.101 | 0.000 | 0.077 | 1.755 | 0.000 | 0.031 | 1.755 |
| 174 | 421.772 | 0.101 | 0.101 | 0.000 | 0.077 | 2.959 | 0.000 | 0.087 | 2.960 |
| 175 | 99.522 | 0.101 | 0.101 | 0.000 | 0.077 | 1.497 | 0.000 | 0.022 | 1.498 |
| 177 | 238.448 | 0.101 | 0.101 | 0.000 | 0.077 | 3.301 | 0.000 | 0.108 | 3.303 |
| 187 | 29.309 | 0.089 | 0.089 | 0.000 | 0.071 | 0.788 | 0.000 | -0.013 | 0.788 |
| 188 | 121.834 | 0.089 | 0.089 | 0.000 | 0.071 | 1.631 | 0.000 | -0.057 | 1.632 |
| 190 | 44.118 | 0.126 | 0.126 | 0.003 | 0.083 | 1.530 | 0.409 | 0.441 | 1.644 |
| 192 | 37.943 | 0.089 | 0.089 | 0.000 | 0.071 | 0.793 | 0.000 | -0.014 | 0.793 |





| 200 | 2346.695 | 0.113 | 0.113 | 0.001 | 0.084 | 8.428 | 0.000 | 3.832 | 9.258 |
|-----|----------|-------|-------|--------|-------|-------|-------|--------|--------|
| 202 | 530.794 | 0.081 | 0.081 | 0.007 | 0.057 | 2.212 | 0.000 | 4.823 | 5.306 |
| 206 | 379.500 | 0.081 | 0.081 | 0.007 | 0.057 | 2.731 | 0.000 | 7.352 | 7.842 |
| 210 | 777.085 | 0.145 | 0.146 | 0.014 | 0.117 | 5.916 | 0.000 | 22.988 | 23.737 |
| 211 | 3155.842 | 0.145 | 0.146 | 0.014 | 0.117 | 7.761 | 0.000 | 39.564 | 40.318 |
| 212 | 319.990 | 0.145 | 0.146 | 0.014 | 0.117 | 4.443 | 0.000 | 12.964 | 13.704 |
| 213 | 54.459 | 0.145 | 0.146 | 0.014 | 0.117 | 3.107 | 0.000 | 6.342 | 7.062 |
| 237 | 519.380 | 0.096 | 0.096 | 0.001 | 0.073 | 3.493 | 0.000 | 1.332 | 3.738 |
| 243 | 595.395 | 0.074 | 0.074 | 0.002 | 0.058 | 2.534 | 0.000 | 1.745 | 3.076 |
| 246 | 370.096 | 0.089 | 0.089 | 0.005 | 0.070 | 2.593 | 0.000 | 4.037 | 4.798 |
| 248 | 780.960 | 0.081 | 0.081 | 0.007 | 0.057 | 3.222 | 0.000 | 10.231 | 10.726 |
| 256 | 14.021 | 0.073 | 0.073 | -0.001 | 0.058 | 0.618 | 0.000 | -0.062 | 0.622 |
| 258 | 12.868 | 0.079 | 0.079 | 0.002 | 0.063 | 0.635 | 0.000 | 0.134 | 0.649 |
| 258 | 21.311 | 0.079 | 0.079 | 0.002 | 0.063 | 0.692 | 0.000 | 0.160 | 0.710 |
| 265 | 92.805 | 0.074 | 0.074 | 0.001 | 0.057 | 1.576 | 0.000 | 0.230 | 1.593 |
| 266 | 119.335 | 0.062 | 0.062 | 0.004 | 0.048 | 1.298 | 0.000 | 1.540 | 2.014 |
| 267 | 475.928 | 0.074 | 0.074 | 0.002 | 0.058 | 1.758 | 0.000 | 0.840 | 1.949 |
| 268 | 1221.917 | 0.074 | 0.074 | 0.002 | 0.058 | 3.240 | 0.000 | 2.853 | 4.317 |
| 269 | 616.709 | 0.074 | 0.074 | 0.002 | 0.058 | 2.713 | 0.000 | 2.000 | 3.370 |
| 270 | 551.152 | 0.120 | 0.120 | 0.006 | 0.096 | 3.273 | 0.000 | 4.730 | 5.752 |
| 271 | 145.884 | 0.120 | 0.120 | 0.006 | 0.096 | 1.850 | 0.000 | 1.510 | 2.388 |
| 273 | 548.626 | 0.120 | 0.120 | 0.006 | 0.096 | 3.244 | 0.000 | 4.646 | 5.667 |
| 277 | 542.262 | 0.145 | 0.146 | 0.014 | 0.117 | 4.043 | 0.000 | 10.738 | 11.474 |
| 279 | 1022.168 | 0.158 | 0.158 | -0.008 | 0.120 | 5.410 | 0.000 | -9.416 | 10.860 |
| 282 | 30.905 | 0.105 | 0.105 | 0.005 | 0.076 | 1.362 | 0.000 | 0.794 | 1.577 |
| 673 | 144.566 | 0.101 | 0.101 | 0.000 | 0.077 | 1.894 | 0.000 | 0.036 | 1.894 |


**Data availability**

The analysed meteorological data are available at the Tyrolean Hydropower Company TIWAG (www.tiwag.at) and at the "Hydrographischer Dienst Vorarlberg" (eHYD: https://ehyd.gv.at). Used orthophotos and single historical aerial images are available at the data services of the Province of Tyrol (www.data.gv.at) and the Federal Office of Metrology and Surveying

(BEV; www.bev.gv.at). The DEM of 2006 was provided by the Province of Tyrol (www.tirol.gv.at). The LiDAR data of 2019 will be publicly available after completion of the SEHAG (SEnsitivity of High Alpine Geosystems to climate change since 1850) research project and can be provided upon request.



**Author contributions**

Planning and conceptualization was done by JR, FH and MB. Responsible for data curation were JR, FH, TH, MA, FF, CR and SBN. The mapping was done by JR and the analyses were performed by JR, MA, FF and SBN. Supervision was provided by FH, TH and MB. The original manuscript was written by JR. FH, TH, MA, FF, CR and MB were involved in reviewing and editing of the manuscript. MB, FH and TH were responsible for funding acquisition.

**Competing interests**

The authors declare that they have no conflict of interest.

**Acknowledgments**

This study is part of the SEHAG (SEnsitivity of High Alpine Geosystems to climate change since 1850) research project, which is financially supported by the German Research Foundation (DFG), the Austrian Science Fund (FWF), the autonomous province of South Tyrol and by the Swiss National Science Foundation (SNF). For providing all the essential data, we would like to thank the Tyrolean Hydropower Company (TIWAG), the Hydrographischer Dienst Vorarlberg, the Federal Office of Metrology and Surveying (BEV) and the Province of Tyrol (Land Tirol). Furthermore, we want to thank the Bezirkshauptmannschaft Imst (especially Mag. Eva Loidhold and Mag. Gudrun Hofmann), the municipality of Umhausen with the Mayor Mag. Jakob Wolf, as well as Mag. Johannes Kostenzer, Dr. Werner Schwarz, Mag. Kathrin Herzer and all residents of Niederthai and Umhausen for supporting the research projects in the Horlachtal. Special thanks to all student assistants who supported our studies.

**Financial support**

This research has been supported by the German Research Foundation (DFG) (grant numbers BE 1118/38-1, BE 1118/39-1, BE 1118/40-1, HA 5740/10-1 and HE 5747/6-1) as well as by the Austrian Science Fund (FWF) (grant number 4062-N29).

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
