# Peer review of "Spatio-temporal analysis of slope-type debris flow activity in Horlachtal, Austria based on orthophotos and LiDAR data since 1947"

_Natural Hazards and Earth System Sciences, 2022_

## Author Comment (AC1)

**Table 3: Calculated parameters of the hydrological catchments of the slope-type debris flows alongside their definitions and dimensions. Correlation of each parameter with the respective debris flow volumes were calculated by Spearman's rho. The p-values represent the significance of the correlations.**

| catchment parameter | index and definition | dimension | Spearman's rho (all) | p-value (all) | Spearman's rho (n > 1) | p-value (n > 1) |
|---|---|---|---|---|---|---|
| planimetric area | A | m² | 0.46 | < 2.2e-16 | 0.39 | 6.1e-12 |
| length | L | m | 0.43 | < 2.2e-16 | 0.35 | 9.6e-10 |
| perimeter | P | m | 0.46 | < 2.2e-16 | 0.38 | 2.3e-11 |
| mean slope | S | degree | 0.02 | 0.74 | -0.04 | 0.50 |
| relief | H | m | 0.43 | < 2.2e-16 | 0.34 | 1.1e-9 |
| Melton ratio | $M=H/\sqrt{A}$ | - | -0.22 | 1.1e-5 | -0.20 | 7.6e-4 |
| relief ratio | R=H/L | - | -0.11 | 0.02 | -0.12 | 0.04 |
| form factor | $F=(A/L)^2$ | m | 0.45 | < 2.2e-16 | 0.38 | 1.1e-11 |
| elongation ratio | $E=(4*A/\pi)/L$ | m | 0.45 | < 2.2e-16 | 0.38 | 1.1e-11 |
| circularity | $C=(4*\pi*A)/P^2$ | - | 0.05 | 0.29 | 0.03 | 0.63 |
| drainage density | D=L/A | $m^{-1}$ | -0.45 | < 2.2e-16 | -0.38 | 1.1e-11 |
| cut density | CD=R/P | - | -0.43 | < 2.2e-16 | -0.35 | 4.2e-10 |

---

## Author Response (AR1)

We would like to thank Reviewer 1 for the valuable input to our manuscript. In this document, we answer step by step to every comment. Reviewer's comments are depicted in black, the authors' answers are written in blue. The line references in the authors' comments refer to the new revised manuscript.

**Reviewer 1**

This manuscript reconstructs debris-flow magnitude and frequency in the Horlachtal, Austria, since 1947. It uses extensive geomorphological mapping using historic and recent orthophotos. The authors show that debris-flow activity in this area was dominated by short-term variations rather than consistent increasing or decreasing trends. Furthermore, the analyses points to local thunderstorms triggering debris flows in the Horlachtal.

In my opinion, this work is strongly relevant for the journal of Natural Hazards and Earth System Sciences. The manuscript is based on a solid and extensive analysis. In total 834 debris flows have been mapped, leading to strong statistics. Furthermore, the manuscript is well-written, although figure presentation may be improved. Below I list a number of suggestions for improvement.

**Main:**

*Transport-limited vs supply-limited hillslope systems*. In lines 101-103 the authors state that the debris flows in Horlachtal occur in transport-limited hillslope systems. However, in the discussion the authors argue that highly active periods affect debris-flow activity in the following years by reducing magnitude and frequency as a result of depleted sediment storages (e.g., lines 472-482). This is a textbook example of supply-limited conditions, and therefore the statements in lines 472-782 and 101-103 are in direct contrast with each other.

**Comments from the authors:**

The debris flows in the Horlachtal are transport-limited in the first place.

In the vast majority of the cases, the debris flow material originates on the one hand from glacial morain material covered with rockfall debris (talus slopes). On the other hand, it originates from rockfall material deposits, which is temporarily stored in the catchments of the debris flows.

If there is a heavy precipitation event or several events within in a short period, the rockfall deposits in the catchments may be emptied. In addition, some debris flow channels are strongly incised into the talus slopes. Thus, those debris flows can no longer mobilise the morain material that easily. Only in these occasional cases, we expect a short-term change from a transport-limited system to a supply-limited system.

We stressed this statement more clearly in the manuscript. Therefore, we added the text to Sect. 2 Study area (lines: 108-111) and we changed Sect. 5.2.1 Frequencies and magnitudes in different periods (lines: 469-472).

*Relation between rainfall magnitude and flow magnitude*. On a related note, in transport-limited systems one would expect a correlation between triggering rainfall magnitude and debris-flow magnitude. In contrast, such a correlation becomes weaker and would typically be absent in supply-limited systems because flow magnitudes also are limited by the volume of sediment available. It would therefore be of interest to compare triggering rainfall to flow magnitudes, or given the data availability perhaps maximum rainfall magnitudes in a given period versus the maximum debris-flow magnitudes in the same period (this should at least be possible for sub-catchment ZT judging from the information in section 5.2.2). It may be needed to normalize by catchment size or another morphometric characteristic of the source catchment as this also affects flow magnitude.

**Comments from the authors:**

We have already done these analyses, which showed no correlation between rainfall magnitude and volume.

However, the debris flow triggering extreme precipitation events are far too local to be able to make a well-founded statement here. We think that even for ZT, the precipitation measuring station is still too far away.

*Catchment morphometry versus flow magnitudes*. In section 4.3 the volume of 404 debris flows is compared to a number of morphometric parameters of their source catchments. A key component of such an analysis is information on how many events are generated from each studied source catchment. If each catchment in the dataset generates multiple flows there is stronger statistics, while if each catchment only produces 1 flow this introduces uncertainty since this one flow may have been relatively small or large. It would be good to elaborate on this in the manuscript.

**Comments from the authors:**

We have included this suggestion into our analyses. If we only take those catchments into account that produced at least two debris flows in the studied period, the sample size is reduced from 404 to 296.

The correlations now are a little bit weaker, but the conclusions gained from the analysis do not change as a result.

We changed Sect. 4.3 Analysis of hydrological catchment parameters accordingly (lines: 317-325).

In addition, we supplemented Tab. 3 with the new statistical data.

*Conclusions*. I suggest to shorten the conclusions and also remove the subsections.

**Comments from the authors:**

Reviewer 2 also made a similar comment. As a result, we have rewritten Sect. 6 Conclusion (lines: 548-564).

**Details:**

Lines 97-98: Please elaborate on how these type 2 debris flows in Zimmermann (1990) or type 1 in Wichmann (2006) and Rieger (1999) are defined, e.g., describe their characteristics.

**Comments from the authors:**

We rephrased the paragraph in Sect. 2 Study area (lines: 101-104) for a better understanding.

Lines 162-163: Two times "The approach".

Line 2019: Parameter should be parameters.

**Comments from the authors:**

Those mistakes were corrected in lines 168 and 225.

Lines: 270-274: "The mapping of the debris flows showed a concentration of these processes in the parallel sub-catchments GT, LT and ZT. As those debris flows show such a different picture when comparing them to the activity in the other sub-catchments, and because of the similarities in the geomorphological and geographical settings, the analyses concerning deposition volumes were carried out exclusively in GT, LT and ZT." It is unclear how the debris flows of sub-catchment GT, LT, and ZT differ from those of the other catchment. As such, this statement raises a lot of questions. Please clarify.

**Comments from the authors:**

There were several reasons for this decision. First, the topographical situation in GT, LT and ZT is very well comparable. Second, the depositions of the HT debris flows are hidden under dense vegetation in most of the cases. A correct and detailed mapping of these depositions (like in GT, LT and ZT) is therefore not possible. Third, the debris flows in FT and WK are by far smaller than those of GT, LT and ZT. As a consequence, we were not able to delineate (or even identify) the FT and WK debris flow depositions. These reasons led us to the conclusion to only use the debris flows in GT, LT and ZT for the volume analyses.

We described these reasons now in more detail in the revised manuscript in Sect. 4.2 Debris flow volumes (lines: 274-280).

Line 300. Include space in "manyevents".

**Comments from the authors:**

The mistake was corrected in line 305.

Lines 297-301. To me the most striking feature in Fig. 8 is the strong increase in flow volume around 1990. Therefore, it would be good to also describe that here.

**Comments from the authors:**

We added a description to this feature in Sect. 4.2 Debris flow volumes (lines: 300-302).

Line 532. Include space in "adetailed".

**Comments from the authors:**

This Section was deleted.

**Figures:**

Overall: In many figures the font sizes are too small and should be enlarged.

Figure 2: It would be more informative to not only plot mean temperatures and precipitation, but rather plot a band indicating the values range. For example, mean +- std or 25-50-75 percentiles.

**Comments from the authors:**

Fig. 2 was supplemented accordingly. We added 25 and 75 percentiles to both temperature and precipitation.

Figure 5: For readability the font size on the axes should be enlarged.

**Comments from the authors:**

We have enlarged the font sizes in Fig. 5.

Figures 6 and 8: Given the unequal intervals of the time slices the left panels of these figures are not informative. I therefore suggest that the authors only present the data on the right panels, and combine debris flow frequency (Fig. 6) and magnitude (Fig. 8) in one figure. Do not denode panels as left and right, but annotate as "a" and "b". In addition, also for these

figures the font size is too small and I suggest enlarging the font. In addition, for the magnitudes it would be beneficial to also include uncertainties with dashed lines.

**Comments from the authors:**

We changed the figures 6 and 8 as suggested. As a result, we created a combined figure (new Fig. 6) and deleted Fig. 8. The numbering of the following figures changed as a consequence.

Figure 12: For comparability it would be better to present the magnitude-frequency curves in panels b and c together in one panel. Also gridlines would help interpretation of the figures. Again, font sizes should be enlarged in this figure.

**Comments from the authors:**

We changed Fig. 12 (new Fig. 11) as suggested.

We would like to thank Reviewer 2 for the valuable input to our manuscript. In this document, we answer step by step to every comment. Reviewer's comments are depicted in black, the authors' answers are written in blue. The line references in the authors' comments refer to the new revised manuscript.

**Reviewer 2**

This manuscript presents a survey of the slope-type debris flow activity in the Horlachtal region of Austria since 1947 based on historical and recent region-wide orthophotos and LiDAR data, with the expectation that the spatial and temporal changes of a debris flow can be reflected through geomorphological changes. The manuscript presents rich debris flow data and conducts extensive data analysis. These works are relatively substantial, which is in line with the interests of the potential readers of NHESS.

Reviewer still have some questions and also some suggestions about the current research and manuscript, the comments can be found in below:

Major comments:

The scientific challenge of the manuscript need to be further sorted out. The author hopes to explore the spatial and temporal distribution characteristics of local slope-type debris flows, however, there seems to be no clear rule or conclusion until the end of the manuscript.

**Comments from the authors:**

The aim of the paper was to analyse slope-type debris flows in a high alpine system in both spatial and temporal ways, in order to better characterize the process dynamics. To be able to make statements about the future development of slope-type debris flows, both spatial and temporal pattern records in the past are required. Part of this study was to establish these records and combine it with precipitation datasets.

We precised the aim of the study in the revised manuscript in Sect. 1 Introduction (lines: 56-66).

However, most of the methods we used (like the geomorphological mapping or the volume analyses) provide both spatial and temporal results. This is why we have taken up this distinction again in the Discussion. In the new rewritten Conclusion, we have seperated conclusions concerning spatial and temporal characteristics as well.

The results indicate that the slope-type debris flow activities in the Horlachtal region show three active periods. However, they seem to be artificially divided. Under this premise,

whether the statistical results of debris flows in different periods, especially the quantity, are in line with the actual situation. In reviewer's opinion, people can get good statistical results they want by adjusting the time interval. Therefore, the basis of three active periods may need clarifications and solid reference.

**Comments from the authors:**

The used method influences the boundaries of the active periods. As a consequence, these boundaries are not chosen randomly, but are determined in advance by the availability of historical aerial image surveys.

The only way to compare these periods with alternating durations is to normalise by the number of years between the periods.

This is described in the manuscript in lines 529-531: "*The calculations of 'debris flows per year' suggest a uniformly distributed debris flows activity throughout the respective epochs, which is far from reality, and hence these calculations should be treated with caution*".

It is therefore due to the methodology that this approach is somewhat problematic in order to be able to delineate the "real" active periods with high accuracy.

Because debris flow triggering rainfall events occur very local, we cannot use the precipitation data of the meteorological station for a better delineation of the active periods.

We integrated new text passages discussing about that topic in Sect. 5.2.1 Frequencies and magnitudes in different periods; Subsection Temporal development of slope-type debris flow activity (lines: 430-435).

In Abstract, authors points out that local thunderstorms are the triggering factors of debris flows. In this manuscript, only very limited words are used to describe this phenomenon. In reveiwer's opinion, the existing materials cannot support this conclusion. Furthermore, this conclusion does not seem to be closely related to the subject of the manuscript, and it is not the main result of the study. Therefore, reviewer does not believe it is appropriate to mention in the Abstract as the main conclusion of the manuscript.

**Comments from the authors:**

We agree with the Reviewer that the thunderstorms as such were no central part of the study. However, we think that during our analyses, we have come across evidence to suggest that the debris flow triggering events in the study area can occur on a very local scale. As described in Sect. 5.1 (lines: 367-374), Figure 5 shows that two neighbouring valleys can display a complete different picture of debris flow activity within the same period.

There are more than one example for this conclusion. Most striking is the high number of debris flows in GT and LT 1990-1997 and the very low activity in ZT in the same timespan. Between 2003 and 2009, however, the debris flow activity in the three sub-catchments happened to be the other way around. Such different debris flow behaviour of neighbouring areas can be seen in even more periods.

We therefore argue that the local scales of debris flow triggering precipitation events are part of the conclusions of this study. But, as suggested by the Reviewer, we cannot adress these events as "thunderstorms". As a consequence, we changed the wording of the Abstract accordingly (lines: 16-19).

The manuscript mainly focuses on the spatiotemporal statistics of debris flows. However, the analysis of the causes of these laws and their physical mechanisms is relatively limited. The susceptibility of debris flow is affected by some important factors such as soil properties and vegetation conditions. In the analysis, the influence of the above factors should be further discussed in combination with the characteristics of the study area.

**Comments from the authors:**

In the vast majority of the cases, the slope-type debris flows in the study area are generated in the hydrological catchments (consisting of bedrock) respectively at the contact zone of the catchments with the talus slope.

In Horlachtal, no trees or higher vegetation grow in the catchments. There is also hardly any soil formation there due to the altitude and the high morphodynamics. If there is any soil formation, then only shallow initial soils.

For this type of debris flows, the literature suggest that the main triggering factors are precipitation events, the morphometrics of the catchments and the slope gradient at the starting zones. We conducted some analyses of the slope gradients at the debris flow starting points in our study area and found that 96 % of them exceed the threshold given in the literature (27°).

In the revised manuscript, we added a whole new subsection within Sect. 5.1 Spatial variability of slope-type debris flows (lines: 385-394). Here, we describe the aforementioned factors for debris flow initiation. In addition, we added specifications on the vegetation of the study area in Sect. 2 Study area (lines: 80-81).

The structure of the manuscript needs to be further streamlined and optimized. For example, "Methodological limitations" are suggested to be placed after the discussion, rather than before each discussion, which will hinder readers' understanding of the research conclusions. In addition, "Conclusions" in the current manuscript need to be modified. It is recommended to refer to other literatures published in NHESS for further simplification to show the insight, impact and implication of current study.

**Comments from the authors:**

Reviewer 1 had similar comments on the conclusion.

As suggested, we restructured the Discussion and rearranged the "Methodological limitations". The Sect. 6 Conclusion was rewritten completely (lines: 548-564).

Minor comments:

It is recommended to further modify the figures:

The font sizes in Figures 5, 6, 8, 9, 10, 11, and 12 are too small, It is recommended to adjust according to the journal requirements;

**Comments from the authors:**

We adjusted font sizes in the mentioned figures to enhance readibility.

For debrs flow volume, uncertainties of the calculations are presented by error bars in Figure 8, so other volume related figures may alos need erro bars?

**Comments from the authors:**

We added error bars for other figures with volume measurements, namely new Fig. 13.

---

## Author Response (AR2)

**Review response**

We would like to thank the Reviewer for the valuable input to our manuscript. In this document, we answer step by step to every comment. Reviewer's comments are depicted in black, the authors' answers are written in red. The line references in the authors' comments refer to the new revised manuscript.

Authors responded most of the comments during the first round. There is no doubt that the data of the manuscript is very comprehensive, and this work is strongly relevant for the journal of Natural Hazards and Earth System Sciences. However, there are still some issues that cause confusion in the article. Some suggestions can be addressed or clarified in revised manuscript:

(1) The author emphasizes in the introduction that the purpose of this manuscript is to discuss the influence of climate change on slope-type debris flow. However, in fact, it is only considered from the perspective of heavy rain, without considering the related indicators of extreme climate conditions including precipitation and temperature (for example, the number of continuous rainy days and extreme temperature, suggesting to refer to the extreme climate index), which could not fully reveal the control effect of climate change on debris flow. On the other hand, debris flow events are not only related to the single rainfall, but also related to the previous rainfall. Clarifications are welcomed here.

The main objective of the manuscript is to analyse and to discuss the development of frequencies and magnitudes of the debris flows in the study area since 1947.

One prominent explanation for possible changes in this development is that the climatic conditions are changing. Because the debris flows in our study area are known to be triggered by high-intensity precipitation events in the vast majority of the cases (Becht, 1995; Becht and Rieger, 1997), the most relevant climatic factor is the number of these events. This is what we refer to in our study by including heavy rainfall data to discuss the temporal development of debris flow frequencies. It is therefore not the main objective of the paper to include multiple climatic parameters.

We changed the respective passage in the Introduction (lines 64-67) to:

*"Because high-intensity rainfall events are decisive for the initiation of debris flows in the study area, this paper aims to analyse the spatial and temporal differences in slope-type debris flow activity in the Horlachtal with the help of temporal high resolution precipitation data. Thus, we want to gain a better understanding of the process behavior throughout the past seven decades and link the results to changes in precipitation patterns due to the changing climate."*

The surface runoff necessary to initiate debris flows in the study area are generated in the bedrock sections of the catchments. Here, it is not relevant whether there was antecedent rainfall or not, as all rainwater drains on the surface and the infiltration capacity does not change. This is why we did not consider antecedent rainfall as a main factor for debris flow initiation in the study area. We clarified that in lines 112-115:

*"In contrast to other types of debris flow systems, the initiation of debris flows on the slopes of the study area is not affected by pre-event conditions like antecedent rainfall, as the necessary runoff is*

*formed in bedrock areas. The most important driving factors for debris flow initiation in Horlachtal are thus high rainfall intensities that generate high peaks of surface runoff."*

(2) Loose material information is an important factor to discuss the formation of debris flow. It is recommended to add the estimation of potential source volume and its connectivity. Also, it is suggested to discuss the influence of earthquake and ice melt water, based on Seismic cataloguing and snow cover datasets.

We already described the type of debris flow material in our study area in lines 109-112:

*"The debris flow material originates on the one hand from glacial morain material covered with rockfall debris on the talus slopes. On the other hand, it emerges from rockfall deposits temporarily stored in the bedrock catchments. The debris flows in Horlachtal occur in transport-limited hillslope systems and are triggered by high-intensity precipitation events of about 20 mm in 30 min (Becht, 1995; Becht & Rieger, 1997)."*

There is no hint that earthquakes in Tyrol influence the debris flow activity. They cannot trigger debris flows as they do not provide the water content necessary to initiate such processes. As explained above, the debris flows on the slopes of the Horlachtal are triggered by high-intensity precipitation events. Antecedent rainfall or meltwater only play, if any, a minor role. In addition, snow melt alone cannot reach the intensities necessary to trigger a debris flow in the study area. So there is still a high-intensity precipitation event necessary for an initiation.

Specific comments:

Page 6, Table2: may need a figure to illustrate the image coverage.

In our opinion, a multitemporal coverage map of the single flight campaigns would be rather confusing to the readers. Therefore, we provided a link to a web service of the Province of Tyrol, which shows all coverages.

Lines 123-124: *"The coverage of the individual campaigns can be viewed in the "Laser- und Luftbildatlas Tirol" of the Province of Tyrol (lba.tirol.gv.at/public/karte.xhtml)."*

Page 8, Line 159: Suggest to add the full name of DoD.

We clarified the naming in the manuscript in line 165.

Page 9, Line 194: Suggest to add the full name of lm(), nls().

We clarified the naming in line 200.

Page 13, Line 280: What is the resolution of Lidar data? Suggest to check the debris flow with volume is not result of data error.

We added information about the spatial resolution of the 2019 LiDAR dataset in lines 156-158:

*"The second LiDAR dataset was recorded during a field campaign of the University of Eichstätt-Ingolstadt in 2019 using a Riegl VUX 1LR integrated in a Riegl VP-1 HeliCopterPod (see riegl.com for details) with a spatial resolution of 13.1 points per m² on average."*

Information about the errors in the debris flow volume measurements based on the DoD is provided in Table A1 (appendix). In all cases, the total error is well below the calculated volumes.

Page 17, Figure 9: Suggest use bar to indicate the number of extreme rainfall events.

The total numbers of days with heavy rainfall events (> 10 mm / 30 min) for both stations are already depicted in Figure 10 as cumulative sums over time. In our opinion, there is no need to add a bar showing similar information in Figure 9.

Page 17, Figure10: It is recommended to label years of rapid growth.
We labelled years with exceptionally many days with a high-intensity event in Figure 10. The new figure looks like:

[Figure]

**Figure 10: Cumulative sums of days with precipitation intensities exceeding 10 mm per 30 minutes for Tschagguns (orange) and Horlachalm (blue). Years with exceptionally many days with a high-intensity event (Tschagguns: four per year; Horlachalm: three per year) are marked.**

Therefore, minor revision is suggested.